# Masked Hard-Attention Transformers Recognize Exactly the Star-Free Languages

**Andy Yang**
University of Notre Dame

**David Chiang**
University of Notre Dame

**Dana Angluin**
Yale University

## Abstract

The expressive power of transformers over inputs of unbounded size can be studied through their ability to recognize classes of formal languages. In this paper, we establish exact characterizations of transformers with hard attention (in which all attention is focused on exactly one position) and attention masking (in which each position only attends to positions on one side). With strict masking (each position cannot attend to itself) and without position embeddings, these transformers are expressively equivalent to linear temporal logic (LTL), which defines exactly the star-free languages. A key technique is the use of Boolean RASP as a convenient intermediate language between transformers and LTL. We then take numerous results known for LTL and apply them to transformers, showing how position embeddings, strict masking, and depth all increase expressive power.

## 1   Introduction

Significant progress has been made in the last few years on characterizing the expressivity of transformers (Vaswani et al., 2017) in terms of well-understood classes of formal languages (Strobl et al., 2024b). Results have been obtained for a wide range of variants of transformers, and nearly all take the form of either upper bounds (transformers recognize only languages in class *C*) or lower bounds (transformers recognize all languages in class *C*). In this paper, we establish *exact* characterizations of transformers with hard attention (in which all attention is focused on exactly one position) and attention masking (in which each position *i* only attends to positions on one side of *i*).

With strict masking (in which each position cannot attend to itself) and without position embeddings, these transformers recognize exactly the class of *star-free* regular languages. The left side of Figure 1 summarizes our results relating masked hard-attention transformers and linear temporal logic (**LTL**), which defines exactly the star-free regular languages.

A key technique in these proofs is the use of **B-RASP**, which, like RASP (Weiss et al., 2021), is a small programming language that compiles into transformers. **B-RASP** is restricted to Boolean values and compiles to masked hard-attention transformers. Additionally, a masked hard-attention transformer can be decompiled back to a **B-RASP** program. We use **B-RASP** as an intermediate language between transformers and **LTL**.

The equivalence of masked hard-attention transformers with **LTL** (and other equivalent characterizations, like counter-free automata and first-order logic) enables us to take numerous results known for **LTL** and apply them to transformers, as shown on the right side of Figure 1:

- Strict future-masked rightmost-hard attention is sufficient; adding past-masked, non-masked, and/or leftmost-hard attention does not increase expressivity (Section 5.1).

- Strict masking is important (Section 5.2); without it, masked hard-attention transformers are less expressive, recognizing only the *stutter-invariant* star-free languages.

38th Conference on Neural Information Processing Systems (NeurIPS 2024).

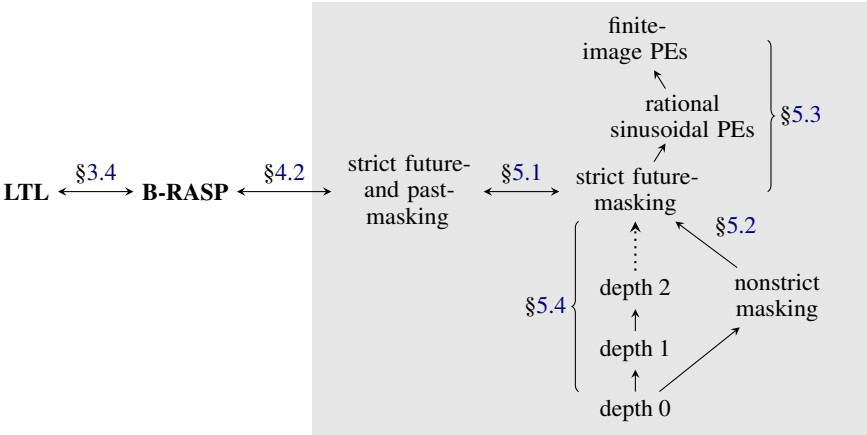

Figure 1: Overview of results in this paper. One-way arrows denote strict inclusion; two-way arrows denote equivalence. PE = position embedding.

- Adding position embeddings increases the class of recognized languages to other well-studied classes (Section 5.3); for example:
  - With rational sinusoidal position embeddings, masked hard-attention transformers recognize exactly the regular languages in $\mathbf{AC}^0$.
  - With arbitrary finite-image position embeddings, they are equivalent to $\mathbf{LTL}[\mathrm{Mon}]$ (linear temporal logic with arbitrary monadic predicates).
- Adding more layers always increases expressive power (Section 5.4).

## 2  Background

### 2.1  Preliminaries

Let $\Sigma$ be a finite alphabet, and let $w = w_1 \cdots w_n$ be an input string of length $n$, where each $w_i \in \Sigma$. Throughout, we assume that $w$ is not empty. We write $\Sigma^+$ for the set of all non-empty strings over $\Sigma$. (We disallow empty strings because several formalisms used here require a designated position where an accept/reject decision appears. Adding a BOS or EOS token not in $\Sigma$ for this purpose would make it possible to handle the empty string.) We write $[n]$ for the set $\{1, \ldots, n\}$.

The *star-free regular languages* are the closure of $\emptyset$, $\{\epsilon\}$, and $\{\sigma\}$ for each $\sigma \in \Sigma$, under the operations of union, concatenation, and complementation. For example:

- $\Sigma^*$ is star-free because $\Sigma^* = \emptyset^{\mathsf{c}}$.
- $(ab)^*$ is star-free because $(ab)^* = (b\Sigma^* \cup \Sigma^*a \cup \Sigma^*aa\Sigma^* \cup \Sigma^*bb\Sigma^*)^{\mathsf{c}}$.
- $(aa)^*$ is regular but not star-free.

This class of languages has several other characterizations, including counter-free automata (Appendix B.5), first-order logic with order (McNaughton and Papert, 1971), and linear temporal logic (Kamp, 1968), which is what we will focus on in this paper.

### 2.2  Transformer variants

The original transformer (Vaswani et al., 2017), designed for machine translation, had both an encoder and a decoder. In practice, both encoder-only models like BERT (Devlin et al., 2019) and decoder-only models like GPT (Brown et al., 2020) are common. Like much previous work on transformer expressivity (e.g. Hahn, 2020), we study an encoder-only setup, where the input is a string and the output is a binary classification; but our results could easily be adapted to a decoder-only setting where the input is a prefix and the output is the next symbol.

The transformers studied here use *unique hard attention* (or simply *hard attention*), in which an attention head focuses all attention on the position with the highest score, with ties broken to the left or right. Although this is different from the soft attention in actual transformers, theoretical studies unavoidably involve models of the real objects of study, and we are using unique-hard attention as a stepping-stone towards understanding real transformers. However, unique-hard attention may be more appropriate than it appears:

- Real transformers are often observed to focus attention on a very small number of positions (Merrill et al., 2021). On Dyck languages, they have been found to learn effectively unique-hard attention in their second layer (Ebrahimi et al., 2020, Figure 1).
- There exist soft-attention transformers that compute parity (Chiang and Cholak, 2022), but in practice, transformers cannot learn parity (Bhattamishra et al., 2020). Unique-hard attention transformers also cannot compute parity (Hahn, 2020), so they are in some sense more realistic.
- Hard attention has occasionally been used in practice in previous research on interpretability (Kinley, 2020) and efficiency (Gupta et al., 2021; Xu et al., 2021).

In this paper, we use *future masking*, in which every position may only attend to positions to its left. This kind of masking is common in decoder-only models and has been studied in encoder-only models as well (Bhattamishra et al., 2020). We also consider *past masking* (Yao et al., 2021).

## 2.3 Previous work

Pérez et al. (2021) show that average-hard attention transformer encoder–decoders, where the decoder runs for a polynomial number of steps before accepting or rejecting a string, recognize all of **P** (that is, all languages decidable by a deterministic Turing machine in polynomial time). Merrill and Sabharwal (2024) prove another version of this result, and further observe that all such transformers are in **P**. This result is the only other exact characterization of any transformer variant that we are aware of.

Hao et al. (2022) show that (non-masked) hard-attention transformer encoders with arbitrary position embeddings have an upper bound of $\mathbf{AC}^0$ (that is, languages defined by circuit families with polynomial size, unbounded fan-in, and bounded depth), and Barceló et al. (2024) show that they have a lower bound of $\mathbf{LTL}[\mathrm{Mon}]$, which is linear temporal logic with all possible monadic numerical predicates. They leave open the question of whether these transformers are equivalent to $\mathbf{LTL}[\mathrm{Mon}]$—a question which, with suitable adjustments, we answer here in the affirmative.

# 3 Boolean RASP

RASP (Weiss et al., 2021) is a programming language intended to help programmers "think like transformers." It has the same basic operations as transformers, but it is easier to compose these operations in RASP than to write transformers by hand. Variants of RASP have been used fruitfully to study transformers' length-generalization capabilities (Zhou et al., 2024) and expressive power (Strobl et al., 2024a; Yang and Chiang, 2024). In this section, we define a version of RASP restricted to Boolean values, which we call Boolean RASP or **B-RASP**. As we will see, it can be compiled into masked hard-attention transformers, and masked hard-attention transformers can be decompiled back into **B-RASP**. We use it as an intermediate language between transformers and **LTL**, and find it more convenient to work with than either of them.

## 3.1 Definition

The input to a **B-RASP** program is a string $w = w_1 \cdots w_n \in \Sigma^+$. There is one type of data, a *Boolean vector*, which is a vector of Boolean values indexed by $i \in [n]$. The *initial* Boolean vectors are $Q_\sigma$ for each $\sigma \in \Sigma$, where $Q_\sigma(i) = 1$ iff $w_i = \sigma$.

A **B-RASP** program is a sequence of operations that compute new Boolean vectors. Although they may have descriptive names, and names may be reused, here, to streamline definitions and proofs, we assume that all the Boolean vectors are numbered consecutively. That is, $P_1, \ldots, P_{|\Sigma|}$ are the initial Boolean vectors $Q_\sigma$ for $\sigma \in \Sigma$, and the Boolean vectors computed by the program are numbered

| input | $\ell$ | $\ell$ | $r$ | $r$ | $\ell$ | $\ell$ | $r$ | $\ell$ | $r$ | $r$ |
|---|---|---|---|---|---|---|---|---|---|---|
| $Q_\ell$ | 1 | 1 | 0 | 0 | 1 | 1 | 0 | 1 | 0 | 0 |
| $Q_r$ | 0 | 0 | 1 | 1 | 0 | 0 | 1 | 0 | 1 | 1 |
| $P_\ell$ | 0 | 1 | 1 | 0 | 0 | 1 | 1 | 0 | 1 | 0 |
| $S_r$ | 0 | 1 | 1 | 0 | 0 | 1 | 0 | 1 | 1 | 0 |
| $I$ | 0 | 1 | 1 | 0 | 0 | 1 | 1 | 1 | 1 | 0 |
| $B_\ell$ | 0 | 1 | 1 | 1 | 0 | 1 | 1 | 1 | 1 | 1 |
| $A_r$ | 1 | 1 | 1 | 0 | 1 | 1 | 1 | 1 | 1 | 0 |
| $C$ | 1 | 1 | 1 | 1 | 1 | 1 | 1 | 1 | 1 | 1 |
| $Y$ | 1 | 1 | 1 | 1 | 1 | 1 | 1 | 1 | 1 | 1 |

| input | $\ell$ | $r$ | $r$ | $\ell$ | $\ell$ | $\ell$ | $r$ | $r$ | $r$ | $\ell$ |
|---|---|---|---|---|---|---|---|---|---|---|
| $Q_\ell$ | 1 | 0 | 0 | 1 | 1 | 1 | 0 | 0 | 0 | 1 |
| $Q_r$ | 0 | 1 | 1 | 0 | 0 | 0 | 1 | 1 | 1 | 0 |
| $P_\ell$ | 0 | 1 | 0 | 0 | 1 | 1 | 1 | 0 | 0 | 0 |
| $S_r$ | 1 | 1 | 0 | 0 | 0 | 1 | 1 | 1 | 0 | 0 |
| $I$ | 1 | 1 | 0 | 0 | 0 | 1 | 1 | 0 | 0 | 0 |
| $B_\ell$ | 0 | 0 | 0 | 0 | 1 | 1 | 1 | 1 | 0 | 0 |
| $A_r$ | 1 | 1 | 0 | 0 | 1 | 1 | 1 | 1 | 0 | 0 |
| $C$ | 1 | 1 | 0 | 0 | 1 | 1 | 1 | 1 | 0 | 0 |
| $Y$ | 0 | 0 | 0 | 0 | 0 | 0 | 0 | 0 | 0 | 0 |

(a) DFA recognizing $L_{1,2}$.  (b) Boolean vectors for membership of string $\ell\ell rr\ell\ell r\ell rr$ in $L_{1,2}$.  (c) Boolean vectors for non-membership of string $\ell rr\ell\ell\ell rrr\ell$ in $L_{1,2}$.

Figure 2: Examples related to $L_{1,2}$ (Dyck-1 of depth 2). The left bracket is $\ell$ and the right bracket is $r$.

starting from $P_{|\Sigma|+1}$ without repetition. After the first $t$ vectors, vector $P_{t+1}$ is computed using one of the following operations.

*Position-wise operations.* $P_{t+1}(i)$ can be be computed by $P_{t+1}(i) := R(i)$, where $R(i)$ is a Boolean combination of zero or more of $\{P_1(i), \ldots, P_t(i)\}$.

*Attention operations.* $P_{t+1}(i)$ can be computed by either of

$$P_{t+1}(i) := \blacktriangleleft_j [M(i, j), S(i, j)] \; V(i, j) : D(i)$$
$$P_{t+1}(i) := \blacktriangleright_j [M(i, j), S(i, j)] \; V(i, j) : D(i)$$

where:

- $M(i, j)$, the *mask predicate*, is one of $M(i, j) = 1$ (no masking), $M(i, j) = (j < i)$ (strict future masking), or $M(i, j) = (j > i)$ (strict past masking).

- $S(i, j)$, the *score predicate*, is a Boolean combination of zero or more atomic formulas from $\{P_1(i), \ldots, P_t(i)\} \cup \{P_1(j), \ldots, P_t(j)\}$.

- $V(i, j)$, the *value predicate*, has the same form as the score predicate.

- $D(i)$, the *default value predicate*, is a Boolean combination of zero or more atomic formulas from $\{P_1(i), \ldots, P_t(i)\}$.

For each $i \in [n]$, let $j_i$ be the minimum (if the operator is $\blacktriangleleft$) or maximum (if $\blacktriangleright$) value of $j \in [n]$ such that $M(i, j) = 1$ and $S(i, j) = 1$. If $j_i$ exists, then $P_{t+1}(i) = V(i, j_i)$. If $j_i$ does not exist, then $P_{t+1}(i) = D(i)$.

If $P$ is a Boolean vector computed by program $\mathcal{P}$, we write $w \models P(i)$ just in case $P(i) = 1$ when $\mathcal{P}$ is run on input string $w$. To make a **B-RASP** program $\mathcal{P}$ recognize a language, one Boolean vector $Y$ is designated the output vector, and position $n$ is designated the output position. Then, the input string $w$ is accepted iff $w \models Y(n)$. To make a **B-RASP** program compute a length-preserving sequence-to-sequence function from $\Sigma^+$ to $\Gamma^+$, we designate a collection of output Boolean vectors $Y_\gamma$ indexed by the symbols $\gamma \in \Gamma$, and consider the output at position $i$ to be $\gamma$ iff $Y_\gamma(i)$ is true.

### 3.2 Example: Dyck-1 of depth 2

As an example, we consider the Dyck language with 1 pair of parentheses, limited to depth 2, or $L_{1,2}$ for short. It is recognized by the DFA in Figure 2a, where $\ell$ and $r$ are left and right brackets. We show how to define this language in **B-RASP**, with a construction very similar to that of Yao et al. (2021).

Consider the input string $\ell\ell rr\ell\ell r\ell rr$, which should be accepted. The basic idea is to identify brackets that are immediately matched ($\underline{\ell\ell rr}\ell\ell\underline{r}\ell rr$), then look at the remaining brackets ($\ell\ell rr\underline{\ell\ell}r\underline{\ell rr}$) to make sure they are matched. We describe the **B-RASP** program for this problem below; the resulting Boolean vectors are shown in Figure 2b.

We first construct Boolean vectors $P_\ell(i)$ and $S_r(i)$ that indicate whether the predecessor (respectively, successor) symbol of $i$ is $\ell$ (respectively, $r$). This is done with attention operations:

$$P_\ell(i) := \blacktriangleright_j [j < i, 1] \ Q_\ell(j) : 0$$
$$S_r(i) := \blacktriangleleft_j [j > i, 1] \ Q_r(j) : 0.$$

Vector $P_\ell(i)$ makes position $i$ attend to the position immediately to its left, and its value predicate $Q_\ell(j)$ tests whether that position has an $\ell$. Vector $S_r$ is similar.

The Boolean vector $I(i)$ indicates whether position $i$ is in a consecutive pair $\ell r$, that is, whether it is *immediately matched*:

$$I(i) := (Q_\ell(i) \wedge S_r(i)) \vee (Q_r(i) \wedge P_\ell(i)).$$

The Boolean vectors $B_\ell(i)$ and $A_r(i)$ test if the symbol before (respectively, after) $i$ that is not immediately matched is $\ell$ (respectively, $r$). Then $C$ checks each position $i$ to see if it is immediately matched, or it has $\ell$ and the following not-immediately-matched symbol is $r$, or it has $r$ and the preceding not-immediately-matched symbol is $\ell$:

$$B_\ell(i) := \blacktriangleright_j [j < i, \neg I(j)] \ Q_\ell(j) : 0$$
$$A_r(i) := \blacktriangleleft_j [j > i, \neg I(j)] \ Q_r(j) : 0$$
$$C(i) := I(i) \vee (Q_\ell(i) \wedge A_r(i)) \vee (Q_r(i) \wedge B_\ell(i)).$$

Finally, the output Boolean vector $Y$ tests if $C(i)$ is true everywhere:

$$Y(i) := \blacktriangleright_j [1, \neg C(j)] \ 0 : 1.$$

Boolean vectors for deciding non-membership of $\ell rr\ell\ell\ell rrr\ell$ in $L_{1,2}$ are shown in Figure 2c. It is straightforward to generalize this technique to recognize Dyck-$k$ of depth $D$ in **B-RASP**.[1] For another example program for an associative recall task, please see Appendix A. A **B-RASP** simulator that allows one to write and run additional examples can be found at `https://b-rasp.github.io/`.

## 3.3   Normal forms

In **B-RASP**, the value predicate $V(i, j)$ depends on both $i$ (the query position) and $j$ (the key/value position), but in actual transformers, it depends on $j$ only. The dependence on $i$ is sometimes convenient, but it does not change expressivity (see Appendix B.1).

The score predicate $S(i, j)$ depends on both $i$ and $j$ in both **B-RASP** and actual transformers. Perhaps surprisingly, in **B-RASP**, it too can be made to depend only on $j$ without reducing expressivity, but as a tradeoff the program may become exponentially larger in size (see Appendix B.2).

## 3.4   Equivalence with linear temporal logic

We prove that **B-RASP** recognizes exactly the star-free languages, by proving that **B-RASP** is equivalent to linear temporal logic. Appendix B.5 gives another proof of the star-free-to-**B-RASP** direction via counter-free automata.

In linear temporal logic or **LTL** (Kamp, 1968), every formula implicitly depends on a single "time" (or position). The atomic formulas are $Q_\sigma$ for every $\sigma \in \Sigma$, and we have the usual connectives $\wedge$, $\vee$, and $\neg$, as well as operators **since** and **until**.[2] For any input string $w = w_1 \cdots w_n$ and position $i \in [n]$,

---

[1]Because we prove in Lemma 21 that **B-RASP** programs can be simulated by masked hard-attention transformers, this result contradicts the claim in Theorem 4.3 (= Theorem C.1) of the paper by Yao et al. (2021); according to a cognizant co-author of that paper, Lemma C.2 in that paper is not true (Peng, 2023)

[2]Other presentations of **LTL** may define non-strict operators **since′** and **until′** (in which $j$ or $k$ can be equal to $i$) and add **previous** and **next** operators. These definitions are expressively equivalent, but the proofs here are more straightforward when defined this way.

we define $w, i \models \phi$ as follows:

$$
\begin{array}{ll}
w, i \models Q_\sigma & \text{if } w_i = \sigma \\
w, i \models \phi_1 \wedge \phi_2 & \text{if } w, i \models \phi_1 \text{ and } w, i \models \phi_2 \\
w, i \models \phi_1 \vee \phi_2 & \text{if } w, i \models \phi_1 \text{ or } w, i \models \phi_2 \\
w, i \models \neg\phi_1 & \text{if } w, i \not\models \phi_1 \\
w, i \models \phi_1 \text{ since } \phi_2 & \text{if for some } j < i, \text{ we have } w, j \models \phi_2, \\
& \text{and for all } k \text{ such that } j < k < i, \text{ we have } w, k \models \phi_1 \\
w, i \models \phi_1 \text{ until } \phi_2 & \text{if for some } j > i, \text{ we have } w, j \models \phi_2, \\
& \text{and for all } k \text{ such that } i < k < j, \text{ we have } w, k \models \phi_1.
\end{array}
$$

To use a formula $\phi$ of **LTL** to define a language over $\Sigma$, for an input string $w \in \Sigma^+$ of length $n$ we designate the last position as the output position, so that $w \in \mathcal{L}(\phi)$ if and only if $w, n \models \phi$.

For example, let $\Sigma = \{a, b, \#\}$ and consider the following formulas:

$$
\begin{aligned}
\phi_1 &= Q_\# \\
\phi_2 &= Q_\# \wedge (Q_b \text{ since } Q_\#) \\
\phi_3 &= Q_\# \wedge (Q_b \text{ since } (Q_\# \wedge (Q_a \text{ since } Q_\#))) \\
\phi_4 &= Q_\# \wedge (Q_b \text{ since } (Q_\# \wedge (Q_a \text{ since } (Q_\# \wedge \neg(0 \text{ since } 1)))))).
\end{aligned}
$$

The formula $\phi_1$ defines the language $\Sigma^*\#$, which contains all and only strings with a # in the last position. The formula $\phi_2$ defines the language $\Sigma^*\#b^*\#$, and $\phi_3$ defines the language $\Sigma^*\#a^*\#b^*\#$. Finally, $\phi_4$ defines the language $\#a^*\#b^*\#$, because $\neg(0 \text{ since } 1)$ is only true at the first position.

**Theorem 1.** *For any formula of* **LTL** *that defines a language* $L \subseteq \Sigma^+$, *there is a* **B-RASP** *program that recognizes* $L$.

*Proof.* See Appendix B.3. This is shown via direct construction. $\quad\square$

**Theorem 2.** *For any* **B-RASP** *program that recognizes a language* $L \subseteq \Sigma^+$, *there is a formula of* **LTL** *that defines* $L$.

*Proof.* See Appendix B.4. We use the unary normal forms (Section 3.3) to facilitate this proof. $\quad\square$

## 4 Masked Hard-Attention Transformers

### 4.1 Definition

A *masked hard-attention transformer layer with width* $d > 0$ is a length-preserving function

$$
\begin{aligned}
& layer \colon (\mathbb{R}^d)^+ \to (\mathbb{R}^d)^+ \\
& (x_1, \ldots, x_n) \mapsto (y_1, \ldots, y_n) \\
& (c_1, \ldots, c_n) = att(x_1, \ldots, x_n) + (x_1, \ldots, x_n) \\
& \qquad y_i = \mathit{ffn}(c_i) + c_i \qquad\qquad\qquad\qquad i = 1, \ldots, n.
\end{aligned}
\tag{1}
$$

The self-attention layer *att* is specified by

- A score function, which is a bilinear function $f_S \colon \mathbb{R}^d \times \mathbb{R}^d \to \mathbb{R}$.
- A mask, which is $M(i, j) = 1$ (no masking), $M(i, j) = (j < i)$ (strict future masking), or $M(i, j) = (i < j)$ (strict past masking).
- A tie-breaking function $C$ to select one element of a finite non-empty set $I \subset \mathbb{N}_+$, which is either $C(I) = \min I$ (choose leftmost position) or $C(I) = \max I$ (choose rightmost position).
- A value function, which is a linear transformation $f_V \colon \mathbb{R}^d \to \mathbb{R}^d$.

The layer works as follows, for each $i \in [n]$. Let

$$
\begin{array}{ll}
U_i = \{j \in [n] \mid M(i, j) = 1\} & \text{unmasked positions} \\
B_i = \{j \in U_i \mid (\forall j' \in U_i)(f_S(x_i, x_{j'}) \le f_S(x_i, x_j))\} & \text{best-scoring unmasked positions}
\end{array}
$$

If $U_i \neq \emptyset$, let $j_i = C(B_i)$ and output $c_i = f_V(x_{j_i})$; but if $U_i = \emptyset$, output $c_i = \mathbf{0}$.

The function *ffn* is a feed-forward neural network with 2 layers and ReLU activations in between.

Then a *masked hard-attention transformer* is a length-preserving function

$$\mathcal{T} \colon \Sigma^+ \to (\mathbb{R}^d)^+$$
$$\mathcal{T} = layer_k \circ \cdots \circ layer_1 \circ emb$$

where $emb \colon \Sigma^+ \to (\mathbb{R}^d)^+$ is a position-wise function (a word embedding), and each $layer_\ell$ is a masked hard-attention transformer layer.

We write $[\mathcal{T}(w)]_i$ for the final activation value at position $i \in [n]$ when $\mathcal{T}$ is run on input $w$. To use $\mathcal{T}$ as a language recognizer, we add an output layer, which linearly projects $[\mathcal{T}(w)]_n$ to a scalar. If the result is nonnegative, we accept $w$; otherwise, we reject. The exact criterion does not matter much, as the transformers we construct only output $+\frac{1}{2}$ or $-\frac{1}{2}$, and could easily be changed to another convention. The language recognized by $\mathcal{T}$ (with the output layer) is the set of strings it accepts.

Our definition above differs from the standard definition (Vaswani et al., 2017) in a few ways besides unique-hard attention, which was discussed above in Section 2.2. Ours lacks layer normalization and position embeddings, but we add them in Sections 4.3 and 5.3, respectively. We only use single-head attention; multi-head attention can be simulated by summing the outputs of multiple single-head attentions, or it can be added to the definition, as in Appendix D.3.1. Our attention masking is strict, but we consider non-strict masking in Section 5.2.

## 4.2 Equivalence with B-RASP

**Theorem 3.** *For any* **B-RASP** *program that recognizes a language $L \subseteq \Sigma^+$, there is a masked hard-attention transformer (with output layer) that recognizes L.*

*Proof.* See Appendix C.1. Attention layers simulate attention operations, and FFNs simulate position-wise operations. □

**Theorem 4.** *For any masked hard-attention transformer (with output layer) that recognizes a language $L \subseteq \Sigma^+$, there is a* **B-RASP** *program that recognizes L.*

*Proof.* See Appendix C.2. To convert a masked hard-attention transformer to **B-RASP**, we first show that all of the intermediate values computed by the transformer are drawn from a finite set and therefore can be represented using $O(1)$ bits.[3] □

## 4.3 Layer normalization

Standard transformers (Vaswani et al., 2017) include layer normalization (Ba et al., 2016), but our definition above does not. Since layer normalization is a position-wise function, the proof of Lemma 24 is unaffected. But the construction of Lemma 21 does need to be modified to circumvent layer normalization (cf. Chiang et al., 2023, Proposition 22). Previously, we used 1 to represent true and 0 to represent false; now, we use a pair of activations to represent a truth value, $(1, 0)$ for true and $(0, 1)$ for false. This ensures that every vector has mean and variance independent of the input $w$, so we can set the parameters of each layer normalization so that it has no effect. (In the proof of Theorem 3, we use a flag to indicate whether there are any unmasked positions or not. This flag already uses the encoding described above, and does not need to be modified.)

## 5 Further Results

In this final section, we leverage results from temporal logic and the equivalences established above to obtain numerous new results for masked hard-attention transformers (and **B-RASP**).

---

[3] Hao et al. (2022) previously proved that hard-attention transformers use $O(\log n)$ bits; the difference is that they assumed arbitrary position embeddings, but we assume either no position embeddings, or position embeddings with finite image (Section 5.3).

## 5.1 Asymmetric attention

Our definitions of both **B-RASP** and masked hard-attention transformers include both leftmost-hard and rightmost-hard attention, and both future and past masking. But we can use the fact that, in **LTL**, if the output is read out only at the last position, it suffices to have only **since** and not **until** (Gabbay et al., 1980) to obtain the following result.

**Theorem 5.** *Both **B-RASP** and transformers with only future-masked rightmost-hard attention recognize exactly the star-free languages.*

*Proof.* Any star-free language can be defined in **LTL** using only **since** (Gabbay et al., 1980), and restricting Theorem 1 to translate from **LTL** with only **since** into **B-RASP** will only use future-masked ▶. Therefore, **B-RASP** with only future-masked ▶ can define any star-free language. Similarly, the translation (Theorem 3) from **B-RASP** with only future-masked ▶ to masked hard-attention transformers only uses future-masked rightmost-hard attention. Therefore, transformers with only future-masked rightmost-hard attention can define any star-free language.  □

Note that this applies only in a setting where we accept or reject strings by looking at the output at the last position. It does not apply to other settings, like transduction (Strobl et al., 2024a).

## 5.2 Non-strict masking

Our definitions of both **B-RASP** and masked hard-attention transformers use strict masking, in which a position cannot attend to itself. Standard transformers, however, use non-strict masking. We can modify the definitions to use *non-strict* masking, that is, $i \leq j$ or $j \leq i$.

Non-strictness is known to reduce expressivity in **LTL** (Peled and Wilke, 1997), so it reduces expressivity in **B-RASP** and masked hard-attention transformers as well. Intuitively, non-strict masked operations are unable to distinguish between consecutive positions that have the same symbol. More formally, a language over $\Sigma$ is called *stutter-invariant*[4] iff for all $u, v \in \Sigma^*$ and $\sigma \in \Sigma$, $u\sigma v \in L$ iff $u\sigma\sigma v \in L$. An example of a language that is stutter-invariant star-free is $(a^+b^+)^*$ (where $\sigma^+$ means "one or more occurrences of $\sigma$"); a language that is star-free but not stutter-invariant is $(ab)^*$.

**Theorem 6.** *Both **B-RASP** and masked hard-attention transformers with only non-strict masking recognize exactly the stutter-invariant star-free languages.*

*Proof.* Peled and Wilke (1997) prove that **LTL** with non-strict **since′** and **until′** recognizes exactly the stutter-invariant star-free languages. The proofs of Theorems 1 and 2 may be adapted to use non-strict temporal operators and non-strict masking. Thus, non-strict **B-RASP** and non-strict **LTL** are equivalent. Similarly, using $j \leq i$ or $j \geq i$ as $M(i, j)$ in the proofs of Theorems 3 and 4, we can show that non-strict masked hard-attention transformers are equivalent to non-strict **B-RASP**.  □

In Section 3.2, we showed how to define $L_{1,2}$, Dyck-1 of depth 2. Bhattamishra et al. (2020, §7.1) find experimentally that $L_{1,2}$ is not learnable by transformers, and they argue that it is not even expressible by transformers (with soft attention, non-strict masking, and no position embeddings). The reason is that while reading the prefix of $\ell$'s at the start of the string, the soft-attention layer computes the same value vector at every position and cannot count the number of occurrences of $\ell$. However, with the addition of a BOS symbol, soft attention can measure what fraction of symbols are $\ell$, overcoming this limitation as observed empirically by Ebrahimi et al. (2020). The similarities between how strict masking in the hard attention setting and the addition of BOS in soft attention both enable positions to be distinguished are notable for future investigation.

## 5.3 Position embeddings

Our definition of a transformer does not, so far, include position embeddings; all information about ordering comes from attention masking. A position embedding is a family of functions $\Theta = (\theta_n)_{n \geq 0}$ where $\theta_n(i)$ is a scalar or vector representation of position $i$ in a string of length $n$. Then the input layer *emb* becomes the sum of a word embedding and a position embedding.

We say that $\Theta$ has *finite image* if $\bigcup_{n \geq 0} \text{Im } \theta_n$ is finite. In general, our results extend to transformers with any position embedding that has finite image. The class of languages recognized may grow, and we give a recipe for characterizing the new class of languages.

---

[4]We thank a reviewer of a previous version of this paper for directing us to the notion of stutter-invariance.

We can add numerical predicates to **LTL** and initial Boolean vectors to **B-RASP** as follows. Let $\Pi = (\pi_n)_{n \geq 0}$ be a family of functions $\pi_n \colon [n] \rightarrow \{0, 1\}$. Then there is an additional predicate symbol $\Pi$ such that for any string $w$ with length $n$,

$$w \models \Pi(i) \text{ iff } \pi_n(i) = 1 \qquad \text{in } \textbf{B-RASP}$$
$$w, i \models \Pi \text{ iff } \pi_n(i) = 1 \qquad \text{in } \textbf{LTL}.$$

For example, if $\text{Mid}_n(i)$ is true iff $n$ is odd and $i = \lceil n/2 \rceil$, then we can define the language $\{\#a^m\#b^m\# \mid m \geq 0\}$ in **LTL**[Mid] as:

$$\phi = Q_\# \wedge (Q_b \textbf{ since } (\text{Mid} \wedge Q_\# \wedge (Q_a \textbf{ since } (Q_\# \wedge \neg(0 \textbf{ since } 1))))).$$

A similar definition could be written in **B-RASP**[Mid].

**Theorem 7.** *Let $\Theta = (\theta_n)_{n \geq 0}$ be a position embedding with finite image. There exists a collection of predicates $\mathcal{P}_\Theta$ such that the following classes of languages are the same:*

- *languages recognized by masked hard-attention transformers with position embedding $\Theta$*

- *languages defined by* **B-RASP**$[\mathcal{P}_\Theta]$

- *languages defined by* **LTL**$[\mathcal{P}_\Theta]$.

*Proof.* See Appendix D.1. ☐

We discuss two important special cases below.

**Sinusoidal position embeddings**    The original transformer (Vaswani et al., 2017) used position embeddings with coordinates of the form $\sin(2\pi f i)$ or $\cos(2\pi f i)$. If the $f$'s are rational (though in the original definition they were not), then the position embeddings form a finite set, so Lemma 22 still holds. For any even $d$, let us define a *rational sinusoidal positional embedding* with $d$ dimensions to be a position embedding $\Theta = (\theta_n)_{n \geq 0}$ where

$$\theta_n(i) = \begin{bmatrix} \sin 2\pi f_1 i & \cos 2\pi f_1 i & \cdots & \sin 2\pi f_{d/2} i & \cos 2\pi f_{d/2} \end{bmatrix}^\top \qquad f_1, \ldots, f_{d/2} \in \mathbb{Q}.$$

**Corollary 8.** *Masked hard-attention transformers with rational sinusoidal position embeddings recognize exactly the regular languages in* $\textbf{AC}^0$ *(that is, regular languages definable by a family of Boolean circuits with polynomial size and constant depth).*

*Proof.* This uses the fact that the regular languages in $\textbf{AC}^0$ are exactly the languages definable in first-order logic with modular predicates (Barrington et al., 1992). See Appendix D.2 for details. ☐

An example of a language that belongs to this class but is not star-free is $(aa)^*$. The classic example of a language that is regular but not in $\textbf{AC}^0$ is PARITY $= \{w \in \{a, b\}^* \mid w \text{ has an odd number of } b\text{'s}\}$ (Furst et al., 1984).

**Arbitrary position embeddings**    Finally, we may consider arbitrary position embeddings, subject to the condition of finite image. The corresponding collection of predicates is the set of all possible monadic predicates, which we call Mon.[5]

**Corollary 9.** *Masked hard-attention transformers that have position embeddings with finite image recognize exactly the languages definable in* **LTL**[Mon].

Barceló et al. (2024) show that any language definable in **LTL**[Mon] can be recognized by a hard-attention transformer without attention masking and with some position embedding (with infinite image), but left the other direction as an open question. Here, by making use of attention masking and restricting position embeddings to those with finite image, we have obtained an exact characterization.

The addition of attention masking appears to be important. With finite image position embeddings but without attention masking, there must be two positions $i$ and $j$ with the same position embedding (by the pigeonhole principle), so an unmasked attention transformer would not be able to distinguish one string with $a$ and $b$ at positions $i$ and $j$ and another string with $a$ and $b$ at positions $j$ and $i$. So no masked hard-attention transformer with finite image position embeddings and unmasked attention can recognize the language $\#a^*\#b^*\#$, but we showed already how to define this language even in **LTL**.

---

[5]Although Barrington et al. (2005) define Mon to be the collection of all monadic predicates without dependence on $n$, other authors (Hao et al., 2022; Barceló et al., 2024) do allow them to depend on $n$.

## 5.4 Depth hierarchy

Finally, we establish that (unlike with feed-forward networks), we can always increase the expressive power of masked hard-attention transformers by adding more self-attention layers. We consider masked hard-attention transformers with only future masking (as is typical in practice) and with only rightmost-hard attention. Other masking and tie-breaking schemes are treated in Appendix D.3. We also add multi-head attention (as is typical in practice).

First, we define depth for all models in this paper. The *layer depth* of a masked hard-attention transformer is the number of attention layers. The *temporal depth* of an **LTL** formula is as follows:

$$\mathrm{dp}(Q_\sigma) = 0 \qquad \mathrm{dp}(\neg\phi) = \mathrm{dp}(\phi)$$
$$\mathrm{dp}(\phi \wedge \psi) = \mathrm{dp}(\phi \vee \psi) = \max(\mathrm{dp}(\phi), \mathrm{dp}(\psi))$$
$$\mathrm{dp}(\phi \textbf{ since } \psi) = \mathrm{dp}(\phi \textbf{ until } \psi) = \max(\mathrm{dp}(\phi), \mathrm{dp}(\psi)) + 1$$

The *attention depth* of a **B-RASP** expression is defined as follows:

$$\mathrm{dp}(Q_\sigma(i)) = 0 \qquad \mathrm{dp}(\neg P(i)) = \mathrm{dp}(P(i))$$
$$\mathrm{dp}(P_1(i) \wedge P_2(i)) = \mathrm{dp}(P_1(i) \vee P_2(i)) = \max(\mathrm{dp}(P_1(i)), \mathrm{dp}(P_2(i))).$$

We then extend this definition to **B-RASP** operations. If $P(i) := \phi(i)$ (a position-wise operation),

$$\mathrm{dp}(P(i)) = \mathrm{dp}(\phi(i)).$$

If $P(i) := \blacktriangleright_j [M(i,j), S(i,j)] \; V(i,j) : D(i)$ or $P(i) := \blacktriangleleft_j [M(i,j), S(i,j)] \; V(i,j) : D(i)$,

$$\mathrm{dp}(P(i)) = \max(\mathrm{dp}(S(i,j)), \mathrm{dp}(V(i,j)), \mathrm{dp}(D(i))) + 1.$$

Finally, the attention depth of a program is the maximum of the attention depths of all of its operations.

Let $\textbf{MUHAT}(\blacktriangleright F)_k$ (respectively, $\textbf{B-RASP}(\blacktriangleright F)_k$) be the languages recognizable by multi-head transformers of depth $k$ (respectively, **B-RASP** programs of depth $k$) using only future-masked rightmost-hard attention. Let $\textbf{LTL}(\textbf{since})_k$ be the languages definable by **LTL** formulas of depth $k$ without **until**.

**Theorem 10.** *For every $k \geq 0$, there is a language $L_{k+1}$ such that no multi-head masked hard-attention transformer of depth $k$ recognizes $L_k$, but a transformer of depth $(k+1)$ does recognize $L_{k+1}$.*

*Proof.* The constructions in the proofs of Theorems 1 and 2 preserve depth, so $\textbf{B-RASP}(\blacktriangleright F)_k = \textbf{LTL}(\textbf{since})_k$. Moreover, by Theorem 4 (shallower version in Appendix C.2), and by Theorem 27 (a depth-preserving version of Theorem 3 found in Appendix D.3), $\textbf{MUHAT}(\blacktriangleright F)_k = \textbf{B-RASP}(\blacktriangleright F)_k$. Finally, Etessami and Wilke (2000) prove that $\textbf{LTL}(\textbf{since})_k \subsetneq \textbf{LTL}(\textbf{since})_{k+1}$. Namely, the classes are separated by $L_{k+1} = \mathrm{STAIR}_{k+1}$, which is the language over $\Sigma = \{a, b, c\}$ of strings which, after deleting $c$'s, contain $a^{k+1}$ as a substring. This gives the following picture:

$$\cdots \subsetneq \quad \textbf{LTL}(\textbf{since})_k \quad \subsetneq \quad \textbf{LTL}(\textbf{since})_{k+1} \quad \subsetneq \cdots$$
$$\parallel \qquad\qquad\qquad \parallel$$
$$\textbf{B-RASP}(\blacktriangleright F)_k \quad \textbf{B-RASP}(\blacktriangleright F)_{k+1}$$
$$\parallel \qquad\qquad\qquad \parallel$$
$$\textbf{MUHAT}(\blacktriangleright F)_k \quad \textbf{MUHAT}(\blacktriangleright F)_{k+1}$$

Therefore, $\textbf{MUHAT}(\blacktriangleright F)_k \subsetneq \textbf{MUHAT}(\blacktriangleright F)_{k+1}$. $\qquad\qquad\qquad\qquad\qquad\qquad\qquad \square$

## 6 Limitations

This work focuses exclusively on masked hard-attention transformers. We discussed the rationale for hard attention in Section 2.2. These results do not apply to softmax-attention transformers, although they demonstrate what kinds of results one might hope to obtain for softmax-attention transformers. Nor do they apply to transformers with unmasked attention.

Finally, our restriction of position embeddings to have finite image, and in particular our restriction of sinusoidal position embeddings to have angles that are rational multiples of $\pi$, does not exactly match the standard definition.

## Acknowledgements

We would like to thank Peter Cholak, Anthony Widjaja Lin, Anand Pillay, and the anonymous reviewers, including the reviewers of a previous version of this paper, for their helpful comments.

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

## A    Additional B-RASP Example: Associative Recall

We consider a variation of the simple associative recall task studied by Friedman et al. (2023) and many others in connection with induction heads and in-context learning (Olsson et al., 2022). Inputs are strings over the alphabet $\{a, b, c, 1, 2, 3\}$, where letters and numbers alternate, starting with a letter and ending with a number, for example $w = a3b2b1a2c1a1c3$. The output alphabet adds the symbol ?. The desired output sequence copies the letters, and for a number at position $i$, if $\sigma$ is the letter at position $i - 1$, then the most recent previous occurrence of $\sigma$ is found, say at position $j$, and the number at position $j + 1$ is output. If there is no previous occurrence of $\sigma$, then ? is output instead. For the given example input $w$, the output should be $y = a?b?b2a3c?a2c1$.

The Boolean vector $P_a$ determines whether there is an $a$ in the preceding position. Similarly, the Boolean vectors $P_b$ and $P_c$ indicate whether there is a $b$ or $c$, respectively, in the preceding position:

$$P_a(i) := \blacktriangleright_j [j < i, 1] \ Q_a(j) : 0$$
$$P_b(i) := \blacktriangleright_j [j < i, 1] \ Q_b(j) : 0$$
$$P_c(i) := \blacktriangleright_j [j < i, 1] \ Q_c(j) : 0.$$

The program has an output Boolean vector $Y_\sigma$ for each symbol $\sigma \in \Sigma$ indicating whether the output symbol is $\sigma$. For $\sigma \in \{1, 2, 3\}$, if position $i$ is preceded by a letter, the output Boolean vector $Y_\sigma$ attends to the most recent position $j$ preceded by the same letter (if any), and returns the value of $Q_\sigma(j)$ for that $j$:

$$Y_1(i) := \blacktriangleright_j [j < i, (P_a(i) \wedge P_a(j)) \vee (P_b(i) \wedge P_b(j)) \vee (P_c(i) \wedge P_c(j))] \ Q_1(j) : 0$$
$$Y_2(i) := \blacktriangleright_j [j < i, (P_a(i) \wedge P_a(j)) \vee (P_b(i) \wedge P_b(j)) \vee (P_c(i) \wedge P_c(j))] \ Q_2(j) : 0$$
$$Y_3(i) := \blacktriangleright_j [j < i, (P_a(i) \wedge P_a(j)) \vee (P_b(i) \wedge P_b(j)) \vee (P_c(i) \wedge P_c(j))] \ Q_3(j) : 0.$$

For $\sigma \in \{a, b, c\}$, the output Boolean vector $Y_\sigma$ is just $Q_\sigma$:

$$Y_a(i) := Q_a(i)$$
$$Y_b(i) := Q_b(i)$$
$$Y_c(i) := Q_c(i).$$

Finally, the output Boolean vector $Y_?$ is true if the position is preceded by a letter but no number was assigned:

$$Y_?(i) := (P_a(i) \vee P_b(i) \vee P_c(i)) \wedge \neg(Y_1(i) \vee Y_2(i) \vee Y_3(i)).$$

The Boolean vectors in the computation for the example string $w$ are shown below.

| input | a | 3 | b | 2 | b | 1 | a | 2 | c | 1 | a | 1 | c | 3 |
|---|---|---|---|---|---|---|---|---|---|---|---|---|---|---|
| $Q_a$ | 1 | 0 | 0 | 0 | 0 | 0 | 1 | 0 | 0 | 0 | 1 | 0 | 0 | 0 |
| $Q_b$ | 0 | 0 | 1 | 0 | 1 | 0 | 0 | 0 | 0 | 0 | 0 | 0 | 0 | 0 |
| $Q_c$ | 0 | 0 | 0 | 0 | 0 | 0 | 0 | 0 | 1 | 0 | 0 | 0 | 1 | 0 |
| $P_a$ | 0 | 1 | 0 | 0 | 0 | 0 | 0 | 1 | 0 | 0 | 0 | 1 | 0 | 0 |
| $P_b$ | 0 | 0 | 0 | 1 | 0 | 1 | 0 | 0 | 0 | 0 | 0 | 0 | 0 | 0 |
| $P_c$ | 0 | 0 | 0 | 0 | 0 | 0 | 0 | 0 | 0 | 1 | 0 | 0 | 0 | 1 |
| $Y_1$ | 0 | 0 | 0 | 0 | 0 | 0 | 0 | 0 | 0 | 0 | 0 | 0 | 0 | 1 |
| $Y_2$ | 0 | 0 | 0 | 0 | 0 | 1 | 0 | 0 | 0 | 0 | 0 | 1 | 0 | 0 |
| $Y_3$ | 0 | 0 | 0 | 0 | 0 | 0 | 0 | 1 | 0 | 0 | 0 | 0 | 0 | 0 |
| $Y_a$ | 1 | 0 | 0 | 0 | 0 | 0 | 1 | 0 | 0 | 0 | 1 | 0 | 0 | 0 |
| $Y_b$ | 0 | 0 | 1 | 0 | 1 | 0 | 0 | 0 | 0 | 0 | 0 | 0 | 0 | 0 |
| $Y_c$ | 0 | 0 | 0 | 0 | 0 | 0 | 0 | 0 | 1 | 0 | 0 | 0 | 1 | 0 |
| $Y_?$ | 0 | 1 | 0 | 1 | 0 | 0 | 0 | 0 | 0 | 1 | 0 | 0 | 0 | 0 |
| output | a | ? | b | ? | b | 2 | a | 3 | c | ? | a | 2 | c | 1 |

# B   Proofs for Section 3 (Boolean RASP)

## B.1   Unary value predicate

**Proposition 11.** *Every* **B-RASP** *program is equivalent to one in which all value predicates* $V(i, j)$ *depend only on* $j$.

*Proof.* This can be seen by the fact that the simulation of **since/until** (Appendix B.3) does not use a value predicate that depends on $i$, but we can show this more directly by induction on the structure of $V(i, j)$. We only show how to handle ▶; the case of ◀ is very similar.

Consider an attention operation with the form

$$P(i) := \blacktriangleright_j \left[ M(i, j), S(i, j) \right] \ V(i, j) : D(i).$$

The base cases are $V(i, j) = B(j)$, for some Boolean vector $B$, which already has the desired form, and $V(i, j) = B(i)$, in which case $P$ is equivalent to

$$A(i) := \blacktriangleright_j \left[ M(i, j), S(i, j) \right] \ 1 : 0$$
$$P(i) := (A(i) \wedge B(i)) \vee (\neg A(i) \wedge D(i)).$$

If $V(i, j) = V_1(i, j) \wedge V_2(i, j)$, then $P(i)$ is equivalent to

$$A(i) := \blacktriangleright_j \left[ M(i, j), S(i, j) \right] \ 1 : 0$$
$$C_1(i) := \blacktriangleright_j \left[ M(i, j), S(i, j) \right] \ V_1(i, j) : 0$$
$$C_2(i) := \blacktriangleright_j \left[ M(i, j), S(i, j) \right] \ V_2(i, j) : 0$$
$$P(i) := (A(i) \wedge C_1(i) \wedge C_2(i)) \vee (\neg A(i) \wedge D(i)).$$

Similarly for disjunction and negation. □

## B.2   Unary score predicate

**Lemma 12.** *Every* **B-RASP** *program is equivalent to one in which all score predicates* $S(i, j)$ *depend only on* $j$.

*Proof.* Again, we only show the case of ▶, as the case of ◀ is very similar. Consider a **B-RASP** attention operation $P$,

$$P(i) := \blacktriangleright_j \left[ M(i, j), S(i, j) \right] \ V(j) : D(i).$$

Observe that

$$S(i, j) = f(A_1(i), \dots, A_{T_A}(i), B_1(j), \dots, B_{T_B}(j))$$

where $f$ is some Boolean function, and the $A_t$ and $B_t$ are **B-RASP** operations. Let $\mathcal{A} = \{A_1, \dots, A_{T_A}\}$, and for each $\chi \subseteq \mathcal{A}$, define an assignment of truth values to the $A_t$,

$$A_t^\chi = \begin{cases} 1 & A_t \in \chi \\ 0 & A_t \notin \chi \end{cases}$$

and use it to define a unary score $S^\chi(j)$ which uses the truth assignment of $\chi$ plugged into the $A_t$:

$$S^\chi(j) = f(A_1^\chi, \dots, A_{T_A}^\chi, B_1(j), \dots, B_{T_B}(j)).$$

Now, $P(i)$ is equivalent to $P'(i)$, where:

$$P^\chi(i) := \blacktriangleright_j \left[ M(i, j), S^\chi(j) \right] \ V(j) : D(i) \qquad \text{for each } \chi \subseteq \mathcal{A}$$

$$P'(i) := \bigvee_{\chi \subseteq \mathcal{A}} \left( P^\chi(i) \wedge \bigwedge_{t \in [T_A]} \left( A_t(i) \leftrightarrow A_t^\chi \right) \right).$$

To see why, observe that for any $i$, there is exactly one truth assignment $\chi_i$ that satisfies $\bigwedge_{t \in [T_A]} \left( A_t(i) \leftrightarrow A_t^\chi \right)$. This $\chi_i$ also makes $S^{\chi_i}(j)$ equivalent to $S(i, j)$, and $P^{\chi_i}(i)$ equivalent to $P(i)$.

Note that each attention operation translates into as many as $2^{T_A + T_B}$ operations. □

## B.3 Proof of Theorem 1 (LTL to B-RASP)

Theorem 1 follows from the following lemma.

**Lemma 13.** *For any formula $\phi$ of **LTL**, there is a **B-RASP** program with a Boolean vector $P_\phi$ such that, for any input $w$ of length $n$ and all $i \in [n]$, we have $w, i \models \phi$ iff $w \models P_\phi(i)$.*

*Proof.* By induction on the structure of the formula $\phi$. We assume that a **B-RASP** program always contains the initial Boolean vectors $Q_\sigma$ for $\sigma \in \Sigma$.

If $\phi = Q_\sigma$: Add operation

$$P_\phi(i) := Q_\sigma(i).$$

If $\phi = \neg \phi_1$: By the induction hypothesis, convert $\phi_1$ to **B-RASP** operations, including one that computes $P_{\phi_1}$. Then add the operation

$$P_\phi(i) := \neg P_{\phi_1}(i).$$

If $\phi = \phi_1 \wedge \phi_2$: By the induction hypothesis, convert $\phi_1$ to **B-RASP** operations, including one that computes $P_{\phi_1}$, then convert $\phi_2$ to **B-RASP** operations, including one that computes $P_{\phi_2}$. Then add operation

$$P_\phi(i) := P_{\phi_1}(i) \wedge P_{\phi_2}(i).$$

If $\phi = \phi_1 \vee \phi_2$: Similar, but add

$$P_\phi(i) := P_{\phi_1}(i) \vee P_{\phi_2}(i).$$

If $\phi = \phi_1$ **since** $\phi_2$: Similar, but add

$$P_\phi(i) := \blacktriangleright_j \left[ j < i, \neg P_{\phi_1}(j) \vee P_{\phi_2}(j) \right] \; P_{\phi_2}(j) : 0.$$

If $\phi = \phi_1$ **until** $\phi_2$: Similar, but add

$$P_\phi(i) := \blacktriangleleft_j \left[ j > i, \neg P_{\phi_1}(j) \vee P_{\phi_2}(j) \right] \; P_{\phi_2}(j) : 0.$$

$\square$

## B.4 Proof of Theorem 2 (B-RASP to LTL)

Theorem 2 follows from the following lemma.

**Lemma 14.** *For any Boolean vector $P$ of a **B-RASP** program $\mathcal{P}$, there is a formula $\phi_P$ of **LTL** such that for any input $w$ of length $n$ and all $i \in [n]$, we have $w \models P(i)$ iff $w, i \models \phi_P$.*

*Proof.* First, by Lemma 12 and Proposition 11 we can rewrite $\mathcal{P}$ to an equivalent program such that every attention operation only uses unary scores and unary values.

Each initial Boolean vector $Q_\sigma(i)$ translates to the atomic formula $Q_\sigma$.

For each operation $P_t(i)$ (for $t > |\Sigma|$), if it is a position-wise operation, that is,

$$P_t(i) := f(P_1(i), \ldots, P_{t-1}(i))$$

where $f$ is a Boolean function, then by the inductive hypothesis, there are **LTL** formulas $\phi_t$ for each $P_t(i)$. Then we convert $P_t(i)$ into $\phi_t = f(\phi_1, \ldots, \phi_{t-1})$.

If $P_t(i)$ is an attention operation, define

$$\begin{aligned}
\textbf{exists}_< \; \phi &= 1 \; \textbf{since} \; \phi \\
\textbf{exists}_> \; \phi &= 1 \; \textbf{until} \; \phi && \text{also known as } \textbf{eventually} \\
\textbf{exists} \; \phi &= (\textbf{exists}_< \; \phi) \vee \phi \vee (\textbf{exists}_> \; \phi) \\
\textbf{rightmost} \; \phi &= \phi \wedge \neg(\textbf{exists}_> \; \phi).
\end{aligned}$$

If $P_t(i)$ uses $\blacktriangleright$ and future masking, that is,

$$P_t(i) := \blacktriangleright_j \left[ j < i, S(j) \right] \; V(j) : D(i)$$

then by the inductive hypothesis, there are **LTL** formulas $\phi_S, \phi_V$ and $\phi_D$ for the corresponding **B-RASP** operations. Then we can convert $P_t(i)$ into the **LTL** formula

$$\phi_t = (\neg\phi_S \text{ since } (\phi_S \wedge \phi_V)) \vee (\neg(\textbf{exists}_< \phi_S) \wedge \phi_D).$$

If $P_t(i)$ uses ▶ and past masking, that is,

$$P_t(i) := \blacktriangleright_j [j > i, S(j)] \ V(j) : D(i)$$

then

$$\phi_t = (\textbf{exists}_> ((\textbf{rightmost } \phi_S) \wedge \phi_V)) \vee (\neg(\textbf{exists}_> \phi_S) \wedge \phi_D).$$

And if $P_t(i)$ uses ▶ with no masking, that is,

$$P_t(i) := \blacktriangleright_j [1, S(j)] \ V(j) : D(i)$$

then

$$\phi_t = (\textbf{exists} ((\textbf{rightmost } \phi_S) \wedge \phi_V)) \vee (\neg(\textbf{exists } \phi_S) \wedge \phi_D).$$

The cases for ◀ are symmetric. □

## B.5 Counter-free automata to B-RASP

In this section, we give an alternative proof of Theorem 1 using the fact that the star-free languages are exactly those recognized by *counter-free automata*.

A *deterministic finite automaton* (DFA) is a tuple $A = (\Sigma, Q, \delta)$, where $\Sigma$ is the input alphabet, $Q$ is the finite set of states, and $\delta \colon Q \times \Sigma \to Q$ is the transition function. A *counter-free automaton* is a DFA in which no string induces a permutation on any subset of $Q$ other than the identity. Schützenberger (1965) proved that the star-free languages are exactly those recognized by counter-free automata.

**Theorem 15.** *For any counter-free DFA that recognizes a language $L \subseteq \Sigma^+$, there is a **B-RASP** program that recognizes $L$.*

A counter-free automaton can be decomposed using Krohn-Rhodes theory into a cascade of *identity-reset automata*, each of which can be simulated in **B-RASP**.

Maler (2010) gives the following automata-theoretic version of the Krohn–Rhodes decomposition theorem, which we explain below.

**Theorem 16** (Maler, 2010, Theorem 3). *For every [deterministic finite] automaton $A$ there exists a cascade decomposition*

$$C = B_1 \circ B_2 \circ \cdots \circ B_k$$

*such that the following are true.*

1. *Each $B_i$ is a permutation–reset automaton.*

2. *There is a homomorphism $\phi$ from $C$ to $A$.*

3. *Any permutation group in some $B_i$ is homomorphic to a subgroup of the transformation semigroup of $A$.*

*The pair $(C, \phi)$ is called a cascade decomposition of $A$.*

If $B_1 = (\Sigma, Q_1, \delta_1)$ and $B_2 = (Q_1 \times \Sigma, Q_2, \delta_2)$ are DFAs, their *cascade product* $C = B_1 \circ B_2$ is the automaton $(\Sigma, Q_1 \times Q_2, \delta)$ such that $\delta(q_1 q_2, \sigma) = (\delta_1(q_1, \sigma), \delta_2(q_2, (q_1, \sigma)))$. (To reduce clutter, we write tuples of states without commas.) The cascade product allows the automaton $B_2$ to see the current state of $B_1$ in deciding what transition to take. We define iterated cascade product inductively by $B_1 \circ B_2 \circ \cdots \circ B_k = (B_1 \circ B_2 \circ \cdots B_{k-1}) \circ B_k$. This allows each $B_i$ to see the current state of all $B_j$ with $j \leq i$ in deciding what transition to take. (For readers more familiar with finite transducers (Mealy machines), $B_1$ and $B_2$ could be thought of as transducers whose transitions are all of the form $\widehat{q} \xrightarrow{\sigma \,:\, (q, \sigma)} \widehat{r}$. Then the cascade product is just composition.)

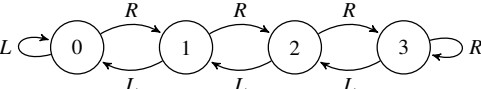

(a) Automaton $A_3$: move right ($R$), move left ($L$).

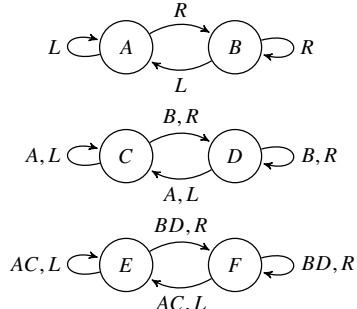

(b) Cascade of identity-reset automata for $A_3$. Omitted transitions are self-loops; for example, inputs $A, R$ and $B, L$ are self-loops on $C$ and $D$.

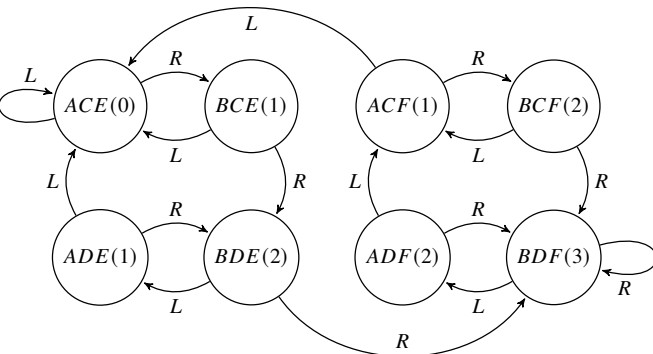

(c) The global automaton of the cascade product in part (b). For state $q$, the number in parentheses is the state $\phi(q)$ in $A_3$.

Figure 3: Example automaton and its cascade decomposition.

In Property (1), a *permutation–reset* automaton is a DFA $A = (\Sigma, Q, \delta)$ such that for every $\sigma \in \Sigma$, the mapping $q \mapsto \delta(q, \sigma)$ is either a permutation (for all $r$, there is a $q$ such that $\delta(q, \sigma) = r$) or constant (there is a $q_\sigma$ such that $\delta(q, \sigma) = q_\sigma$ for all $q$).

Property (2) says that there is a mapping $\phi$ from the states of $C$ to the states of $A$ such that for any state $q$ of $C$, we have $\phi(\delta_C(q, \sigma)) = \delta_A(\phi(q), \sigma)$.

Property (3) implies that if the automaton $A$ is counter-free, then all of the automata $B_i$ in the cascade decomposition are identity–reset automata. An *identity–reset automaton* is a DFA $A = (\Sigma, Q, \delta)$ such that for every $\sigma \in \Sigma$, the mapping $q \mapsto \delta(q, \sigma)$ is either the identity ($\delta(q, \sigma) = q$ for all $q$) or constant (there is a $q_\sigma$ such that $\delta(q, \sigma) = q_\sigma$ for all $q$).

For example, consider the automaton $A_3$ shown in Figure 3a. A decomposition of this automaton into a cascade product of three identity–reset automata is shown in Figure 3b. The global automaton derived from the decomposition is shown in Figure 3c with the homomorphism $\phi$ to states of $A_3$.

Let $A = (\Sigma, Q, \delta)$ be a DFA and $s \in Q$. For a string $w_1 \cdots w_n \in \Sigma^*$, the sequence of states traversed by $A$ from state $s$ on this input is $q_0, \ldots, q_n$, where $q_0 = s$ and for each $k$, $q_{k+1} = \delta(q_k, w_k)$. A **B-RASP** program $p$ *simulates* $A$ started in state $s$ iff for every input word $w \in \Sigma^*$, the output Boolean vectors of $p$ on input $w$ encode the sequence of states traversed by $A$ from state $s$ on input $w$. The state at position $i$ is the state before the symbol at position $i$ is read.

**Lemma 17.** *Let* $B = (\Sigma, Q, \delta)$ *be any identity–reset automaton, and let* $s \in Q$ *be a start state. There exists a* **B-RASP** *program* $\mathcal{P}_B$ *that simulates* $B$ *started in state* $s$.

*Proof.* For each state $r \in Q$, let $R_r \subseteq \Sigma$ be the state of symbols that reset to $r$ (that is, $\delta(q, \sigma) = r$ for all $q \in Q$). Let $R = \bigcup_{r \in Q} R_r$. To determine if $B$ is in state $q \neq s$ before reading $w_i$, it is sufficient to attend to the closest position $j < i$ that contains a symbol from $R$, if any. If $j$ exists and $w_j \in R_q$, then $B$ is in state $q$ at position $i$. Otherwise, it is not.

The case of state $s$ is slightly different. In the case that there is no position $j < i$ that contains a symbol from $R$, then $B$ never left the initial state $s$, so it is still in state $s$ at position $i$.

In **B-RASP**, we can define a Boolean vector $B_q(i)$, which is true iff $B$ is in state $q$ at position $i$:

$$B_q(i) := \blacktriangleright_j \left[ j < i, \bigvee_{\sigma \in R} Q_\sigma(j) \right] \bigvee_{\sigma \in R_q} Q_\sigma(j) : 0 \quad \text{for } q \neq s$$

$$B_s(i) := \blacktriangleright_j \left[ j < i, \bigvee_{\sigma \in R} Q_\sigma(j) \right] \bigvee_{\sigma \in R_s} Q_\sigma(j)) : 1. \qquad \square$$

**Lemma 18.** *Let $B_1 = (\Sigma, Q_1, \delta_1)$ be a DFA that can be simulated from state $s_1$ by a **B-RASP** program $\mathcal{P}_{B_1}$. Let $B_2 = (Q_1 \times \Sigma, Q_2, \delta_2)$ be an identity–reset automaton and let $C = B_1 \circ B_2$. Then there is a **B-RASP** program $\mathcal{P}_C$ that simulates $C$ started in state $(s_1, s_2)$ for an arbitrary $s_2 \in Q_2$.*

*Proof.* Let $B_{1,q}(i)$ be predicates that test whether $B_1$ is in state $q$ at position $i$ started in state $s_1$, and let $B_{2,q}(i)$ be predicates that test whether $B_2$ is in state $q$ at position $i$ started in state $s_2$ (by Lemma 17). Define

$$Q'_{(q,\sigma)}(i) := B_{1,q}(i) \wedge Q_\sigma(i) \qquad\qquad q \in Q_1, \sigma \in \Sigma$$

and for every line in the definition of the $B_{2,q}$, replace every occurrence of $Q_{(q,\sigma)}$ with $Q'_{(q,\sigma)}$. This yields the definition of new predicates $B'_{2,q}$. Then we can define predicates $C_{(q,r)}(i)$ that test whether $C = B_1 \circ B_2$ is in state $(q,r)$ at position $i$:

$$C_{(q,r)}(i) := B_{1,q}(i) \wedge B'_{2,r}(i) \qquad\qquad q \in Q_1, r \in Q_2. \qquad \square$$

By induction on $k$ we have the following.

**Lemma 19.** *Let $C = B_1 \circ B_2 \circ \cdots \circ B_k$ be a cascade product such that each $B_i$ is an identity–reset automaton, and $s_i$ is a state of $B_i$. Then there is a **B-RASP** program $\mathcal{P}_C$ that can simulate $C$ from state $(s_1, s_2, \ldots, s_k)$.*

If we add instructions to the program in Lemma 19 that compute the homomorphism $\phi$ (from property (2) of Theorem 16) from the states of the cascade product $C$ to the automaton $A$:

$$A_r(i) := \bigvee_{\substack{q \in Q \\ \phi(q)=r}} C_q(i)$$

then we get a **B-RASP** program $\mathcal{P}_A$ that simulates $A$ started in state $s$.

Finally, we add to this program position-wise operations $Y_q(i)$ that decide whether $A$ started in state $s$ ends up in state $q$ *after* reading the symbol at position $i$:

$$Y_r(i) := \bigvee_{\substack{q \in Q \\ \sigma \in \Sigma \\ \delta(q,\sigma)=r}} (A_q(i) \wedge Q_\sigma(i)).$$

If $f$ is the final state of $A$, then let $Y_f$ be the output vector of the program. Since $Y_f(n) = 1$ if and only if $A$ accepts $w$, this completes the proof of Theorem 15.

## C   Proofs for Section 4 (Masked Hard-Attention Transformers)

### C.1   Proof of Theorem 3 (B-RASP to masked hard-attention transformers)

We will make use of the following lemma repeatedly:

**Lemma 20.** *Any function* $f \colon \{0,1\}^d \to \{0,1\}^d$ *can be computed by a two-layer FFN with ReLU activation functions.*

*Proof.* Any Boolean function can be written in *full disjunctive normal form* (DNF), which is a disjunction of conjunctions, and each conjunction is a conjunction of one positive or negative literal for each of the arguments, so that at most one conjunction is 1 for any assignment.

Each component of $f$ can be put into full DNF and computed by a two-layer FFN with ReLU activation functions. The first layer computes all the negations and conjunctions, using the fact that for any Boolean values $b_1, b_2, \ldots, b_m$ we have

$$\neg b_1 = 1 - b_1$$
$$b_1 \wedge b_2 \wedge \ldots \wedge b_m = \mathrm{ReLU}(b_1 + b_2 + \ldots + b_m - (m-1)).$$

The second layer computes the disjunction simply by adding the values of the conjunctions. $\qquad\square$

Let $\mathcal{P}$ be a **B-RASP** program with Boolean vectors $P_1, \ldots, P_T$. We say that a masked hard-attention transformer $\mathcal{T}$ with width $d \geq T$ *simulates* $\mathcal{P}$ iff for every input $w \in \Sigma^+$ with length $n$, we have, for all $i \in [n]$ and $t \in [T]$,

$$[\mathcal{T}(w)]_{i,t} = \begin{cases} 1 & \text{if } w \models P_t(i) \\ 0 & \text{otherwise.} \end{cases}$$

Theorem 3 follows from the following lemma:

**Lemma 21.** *Let $\mathcal{P}$ be a **B-RASP** program. There exists a masked hard-attention transformer $\mathcal{T}_\mathcal{P}$ that simulates $\mathcal{P}$.*

*Proof.* We prove that the first $t$ operations of $\mathcal{P}$ can be simulated by a masked hard-attention transformer, by induction on $t$.

The base case is $t = |\Sigma|$. For $c \in 1, \ldots, |\Sigma|$, let $\sigma_c$ be the $c$-th symbol in $\Sigma$. Let $emb(\sigma_c)$ be the one-hot vector with $[emb(\sigma_c)]_c = 1$.

For the inductive step, assume that the Boolean vectors $P_1, \ldots, P_t$ can be simulated by a masked hard-attention transformer. We want to show that $P_{t+1}$ can be simulated as well.

If $P_{t+1}(i)$ is a Boolean combination of $\{P_1(i), \ldots, P_t(i)\}$, it can be computed by a two-layer FFN by Lemma 20.

The most important case is if $P_{t+1}(i)$ is an attention operation, either of

$$P_{t+1}(i) := \blacktriangleleft_j [M(i,j), S(i,j)] \; V(i,j) : D(i)$$
$$P_{t+1}(i) := \blacktriangleright_j [M(i,j), S(i,j)] \; V(i,j) : D(i).$$

We only show the $\blacktriangleright$ case; $\blacktriangleleft$ is similar.

We need to slightly modify the value predicate:

$$V'(i,j) = (S(i,j) \wedge V(i,j)) \vee (\neg S(i,j) \wedge D(i)).$$

This does not change the behavior of $P_{t+1}$ (because $S(i,j)$ is always true when evaluating the value predicate), but it will preserve the correct behavior in a transformer, where attention attends to the leftmost *maximum* score. By Proposition 11, the attention operation can be rewritten so that that the value predicate depends only on $j$. So, without loss of generality, assume that there is a Boolean function $g$ such that

$$V'(i,j) = g(P_1(j), \ldots, P_t(j)).$$

The score predicate $S(i,j)$ can be written in full DNF in terms of the $P_\ell(i)$ and $P_\ell(j)$. We separate each conjunction of $S(i,j)$ into a conjunction of literals depending on $i$ and a conjunction of literals depending on $j$. Thus, there is a collection of Boolean functions $\alpha_\ell$ and $\beta_\ell$ such that

$$S(i,j) = \bigvee_{\ell=1}^{m} (\alpha_\ell(P_1(i), \ldots, P_t(i)) \wedge \beta_\ell(P_1(j), \ldots, P_t(j))).$$

We construct two layers, as follows. For brevity, we write

$$\vec{P}(i) = \begin{bmatrix} P_1(i) \\ \vdots \\ P_t(i) \end{bmatrix}$$

$$\vec{\alpha}(v_1,\ldots,v_t) = \begin{bmatrix} \alpha_1(v_1,\ldots,v_t) \\ \vdots \\ \alpha_m(v_1,\ldots,v_t) \end{bmatrix} \qquad \vec{\beta}(v_1,\ldots,v_t) = \begin{bmatrix} \beta_1(v_1,\ldots,v_t) \\ \vdots \\ \beta_m(v_1,\ldots,v_t). \end{bmatrix}$$

and for any $m$ we write $\mathbf{0}^m$ for the $m$-dimensional zero vector.

Assume that the input to the first layer is

$$x_i^{(1)} = \begin{bmatrix} \vec{P}(i) \\ 0 \\ \mathbf{0}^{T-t-1} \\ \vdots \\ \mathbf{0}^m \\ \mathbf{0}^m \\ 0 \\ 0 \\ 0 \\ 0 \\ \vdots \end{bmatrix} \begin{array}{l} \text{Boolean vectors already computed} \\ \text{Boolean vector to be computed} \\ \text{Boolean vectors not yet computed} \\ \\ \left.\vphantom{\begin{array}{c}1\\1\\1\\1\\1\\1\end{array}}\right\} \text{scratch space} \end{array}$$

The first self-attention has value vectors set to $\mathbf{0}$, so that the residual connection computes the identity function ($c_i^{(1)} = x_i^{(1)}$), and the first position-wise FFN can be constructed, by Lemma 20, so that

$$x_i^{(2)} = f_P^{(1)}\left(c_i^{(1)}\right) + c_i^{(1)}$$

$$= \begin{bmatrix} \vec{P}(i) \\ 0 \\ \mathbf{0}^{T-t-1} \\ \vdots \\ \vec{\alpha}(P_1(i),\ldots,P_t(i)) \\ \vec{\beta}(P_1(i),\ldots,P_t(i)) \\ 0 \\ 1 \\ g(P_1(i),\ldots,P_t(i)) \\ 0 \\ \vdots \end{bmatrix} \begin{array}{l} \\ \\ \\ \\ \text{query} \\ \text{key} \\ \\ \left.\vphantom{\begin{array}{c}1\\1\end{array}}\right\} \text{set default flag to true} \\ \text{value} \end{array} \qquad (2)$$

In the second layer, the self-attention has mask $M$. The score function is

$$f_S^{(2)}\left(x_i^{(2)}, x_j^{(2)}\right) = \left(x_i^{(2)}\right)^{\top} W^S x_j^{(2)}$$

$$= \begin{bmatrix} \vdots \\ \vec{\alpha}(P_1(i),\ldots,P_t(i)) \\ \vec{\beta}(P_1(i),\ldots,P_t(i)) \\ \vdots \end{bmatrix}^{\top} \begin{bmatrix} \ddots & & \\ & \mathbf{0}^{m\times m} \quad \mathbf{I}^{m\times m} & \\ & \mathbf{0}^{m\times m} \quad \mathbf{0}^{m\times m} & \\ & & \ddots \end{bmatrix} \begin{bmatrix} \vdots \\ \vec{\alpha}(P_1(j),\ldots,P_t(j)) \\ \vec{\beta}(P_1(j),\ldots,P_t(j)) \\ \vdots \end{bmatrix}$$

$$= \sum_{\ell=1}^{m} \alpha_\ell(P_1(i),\ldots,P_t(i)) \, \beta_\ell(P_1(j),\ldots,P_t(j))$$

$$= S(i,j)$$

where $\mathbf{0}^{m \times m}$ and $\mathbf{I}^{m \times m}$ are the $m \times m$ zero and identity matrices.

The value function $f_V^{(2)}$ is such that for any $j \in [n]$,

$$
f_V^{(2)}\left(x_j^{(2)}\right) =
\begin{bmatrix}
\mathbf{0}^t \\
0 \\
\mathbf{0}^{T-t-1} \\
\vdots \\
\mathbf{0}^m \\
\mathbf{0}^m \\
1 \\
-1 \\
0 \\
g(P_1(j), \ldots, P_t(j)) \\
\vdots
\end{bmatrix}
\begin{array}{l}
\left.\vphantom{\begin{matrix}1\\-1\\0\end{matrix}}\right\} \text{change default flag to false} \\[1.5em]
\text{value}
\end{array}
$$

So the output of the second self-attention, after the residual connection, is as follows.

If $i > 1$:

$$
c_i^{(2)} = f_V^{(2)}\left(x_{j_i}^{(2)}\right) + x_i^{(2)}
$$

$$
=
\begin{bmatrix}
\vec{P}(i) \\
0 \\
\mathbf{0}^{T-t-1} \\
\vdots \\
\vec{\alpha}(P_1(i), \ldots, P_t(i)) \\
\vec{\beta}(P_1(i), \ldots, P_t(i)) \\
1 \\
0 \\
g(P_1(i), \ldots, P_t(i)) \\
g(P_1(j_i), \ldots, P_t(j_i)) \\
\vdots
\end{bmatrix}
\begin{array}{l}
\\[7em]
\left.\vphantom{\begin{matrix}1\\0\end{matrix}}\right\} \text{default flag (false)} \\[1em]
\text{attended-to value}
\end{array}
$$

If $i = 1$:

$$
c_i^{(2)} = \mathbf{0} + x_i^{(2)}
$$

$$
=
\begin{bmatrix}
\vec{P}(i) \\
0 \\
\mathbf{0}^{T-t-1} \\
\vdots \\
\vec{\alpha}(P_1(i), \ldots, P_t(i)) \\
\vec{\beta}(P_1(i), \ldots, P_t(i)) \\
0 \\
1 \\
g(P_1(i), \ldots, P_t(i)) \\
0 \\
\vdots
\end{bmatrix}
\begin{array}{l}
\\[7em]
\left.\vphantom{\begin{matrix}0\\1\end{matrix}}\right\} \text{default flag (true)} \\[1em]
\text{no value}
\end{array}
$$

The second feed-forward network $f_P^{(2)}$ checks the default flag. If it is $\left[\begin{smallmatrix}1\\0\end{smallmatrix}\right]$, it copies the attended-to value $g(P_1(j_i), \ldots P_t(j_i))$ to the $(t+1)$-st coordinate. If it is $\left[\begin{smallmatrix}0\\1\end{smallmatrix}\right]$, it computes $D(i)$ (using Lemma 20) in the $(t+1)$-st coordinate. Thus, the output after the residual connection is as follows.

If $i > 1$:

$$
y_i^{(2)} = f_P^{(2)}\left(c_i^{(2)}\right) + c_i^{(2)}
$$

$$
=
\begin{bmatrix}
\vec{P}(i) \\
g(P_1(j_i), \ldots, P_t(j_i)) \\
\mathbf{0}^{T-t-1} \\
\vdots \\
\vec{\alpha}(P_1(i), \ldots, P_t(i)) \\
\vec{\beta}(P_1(i), \ldots, P_t(i)) \\
1 \\
0 \\
g(P_1(i), \ldots, P_t(i)) \\
g(P_1(j_i), \ldots, P_t(j_i)) \\
\vdots
\end{bmatrix}
\begin{array}{l}
\text{answer} \\[20em]
\end{array}
$$

If $i = 1$:

$$
y_i^{(2)} = f_P^{(2)}\left(c_i^{(2)}\right) + c_i^{(2)}
$$

$$
=
\begin{bmatrix}
\vec{P}(i) \\
D(i) \\
\mathbf{0}^{T-t-1} \\
\vdots \\
\vec{\alpha}(P_1(i), \ldots, P_t(i)) \\
\vec{\beta}(P_1(i), \ldots, P_t(i)) \\
0 \\
1 \\
g(P_1(i), \ldots, P_t(i)) \\
0 \\
\vdots
\end{bmatrix}
\begin{array}{l}
\text{answer} \\[20em]
\end{array}
$$

In either case, the $(t+1)$-st coordinate is now 1 if $w \models P_{t+1}(i)$ and 0 otherwise. $\qquad\square$

The last step is to construct the output layer, which simply projects the final-layer activation vectors down to the coordinate that simulates $Y$ and subtracts $\frac{1}{2}$. This completes the proof of Theorem 3.

## C.2 Proof of Theorem 4 (masked hard-attention transformers to B-RASP)

The key to the translation from masked hard-attention transformers to **B-RASP** is the following lemma:

**Lemma 22.** *Let $\mathcal{T}$ be a masked hard-attention transformer. There is a finite set $\mathbb{F} \subseteq \mathbb{R}$ such that for all input strings $w$, all the attention scores and activations computed by $\mathcal{T}$ on input $w$ belong to $\mathbb{F}$.*

*Proof.* We prove that regardless of the input, layer $k$ has at most $(|\Sigma| + 1)^{2^k} - 1$ different possible output activation vectors, by induction on $k$. The base case is just the embedding function. Since there are no position embeddings, the embedding at position $i$ is determined entirely by $w_i$, so there are at most $|\Sigma| \leq (|\Sigma| + 1)^{2^k} - 1$ possible activation vectors.

Assume that $\mathcal{T}$ has $(k + 1)$ layers and that layer $k$ has at most $(|\Sigma| + 1)^{2^k} - 1$ possible activation vectors. The self-attention's output at position $i$ depends only on two vectors: (1) layer $k$'s output at position $i$ (because of the residual connection) and (2) either layer $k$'s output at position $j_i$ (the position that $i$ attends to) or $\mathbf{0}$ (if there are no unmasked positions). Thus the number of possible activation vectors that the self-attention can output is at most

$$\left((|\Sigma| + 1)^{2^k} - 1\right)(|\Sigma| + 1)^{2^k} = \left((|\Sigma| + 1)^{2^k}\right)^2 - (|\Sigma| + 1)^{2^k}$$

$$= (|\Sigma| + 1)^{2^{k+1}} - (|\Sigma| + 1)^{2^k}$$

$$\leq (|\Sigma| + 1)^{2^{k+1}} - 1.$$

And the number of possible activation vectors that the position-wise FFN can output is also at most $(|\Sigma| + 1)^{2^{k+1}} - 1$.

As for the attention scores, at layer $(k + 1)$, every attention score depends on two activation vectors from layer $k$, so there are at most $\left((|\Sigma| + 1)^{2^k} - 1\right)^2 \leq (|\Sigma| + 1)^{2^k} - 1$ possible scores.

Then $\mathbb{F}$ is the union over all layers of the possible attention scores and components of the possible activation vectors. $\square$

So any attention score or component of an activation vector can be represented using $B = \lceil \log_2 |\mathbb{F}| \rceil$ bits. Define a mapping $\langle \cdot \rangle \colon \mathbb{F} \to \{0, \ldots, 2^B - 1\}$ such that $u < v$ iff $\langle u \rangle < \langle v \rangle$, and write $\langle v \rangle_b$ for the bit of $\langle v \rangle$ with place value $2^b$.

**Lemma 23.** *For any function $f \colon \mathbb{F} \to \mathbb{F}$, there are Boolean formulas $\phi_b(x_1, \ldots, x_B)$ for $b \in [B]$ such that for any $x \in \mathbb{F}$, $\phi_b(\langle x \rangle_1, \ldots, \langle x \rangle_B)$ holds iff $\langle f(x) \rangle_b = 1$.*

*Proof.* One way to define $\phi_b$ is:

$$\phi_b(x_1, \ldots, x_B) = \bigvee_{\substack{x \in \mathbb{F} \\ \langle f(x) \rangle_b = 1}} \left( \bigwedge_{\substack{b' \in [B] \\ \langle x \rangle_{b'} = 1}} x_{b'} \wedge \bigwedge_{\substack{b' \in [B] \\ \langle x \rangle_{b'} = 0}} \neg x_{b'} \right).$$

Depending on the mapping $\langle \cdot \rangle$, more efficient definitions may be possible. $\square$

Hopefully, it is clear how to generalize this lemma to functions $\mathbb{F}^d \times \mathbb{F}^d \to \{0, 1\}$ or $\mathbb{F}^d \to \mathbb{F}^d$.

Next, we prove Theorem 4. Let $\mathcal{T}$ be a masked hard-attention transformer with width $d$. Let $B$ be the number of bits needed to store $\mathcal{T}$'s activation vector components and attention scores, by Lemma 22. A **B-RASP** program $\mathcal{P}$ *simulates* $\mathcal{T}$ if in $\mathcal{P}$ there are Boolean vectors $Y_{c,b}$ for $c \in [d]$ and $0 \leq b < B$ such that for any input $w \in \Sigma^+$ of length $n$, for all $i \in [n]$, $c \in [d]$, and $0 \leq b < B$, we have $w \models Y_{c,b}(i)$ iff $\langle [\mathcal{T}(w)]_{i,c} \rangle_b = 1$.

**Lemma 24.** *For any masked hard-attention transformer $\mathcal{T}$, there is a **B-RASP** program $\mathcal{P}_T$ that simulates $\mathcal{T}$.*

*Proof.* We proceed by induction on the depth of $\mathcal{T}$. The base case is the input embedding function *emb*, which is simulated by Boolean vectors for $c \in [d]$ and $0 \leq b < B$:

$$\text{Emb}_{c,b}(i) := \bigwedge_{\sigma \in \Sigma} \left( Q_\sigma(i) \to \langle [emb(\sigma)]_c \rangle_b \right).$$

Assume that the first $k$ layers of $\mathcal{T}$ are simulated by a program $\mathcal{P}$. We extend $\mathcal{P}$ to simulate layer $(k + 1)$ as follows.

If the self-attention uses rightmost-hard attention with mask $M(i, j)$, assume (by Lemma 23) that the score function $f_S(i, j)$ has been converted to Boolean expressions $S'_b(i, j)$ for the $b$-th bit of the score for positions $i$ and $j$, and the value function $f_V(j)$ has been converted to Boolean expressions $V'_{c,b}(j)$ for the $b$-bit of the $c$-th coordinate of the value.

We give two translations. The first version has depth 1, which is important in Section 5.4. The second version is deeper in general, but much smaller.

*Shallower version:*

Because $\mathbb{F}$ is finite, by Lemma 23 we can define, for all $v \in \mathbb{F}$, predicates
$$S_v(i, j) \text{ just in case } f_S(i, j) = v$$
$$S_{>v}(i, j) \text{ just in case } f_S(i, j) > v.$$
Then for each $v \in \mathbb{F}$, add operations for $\mathrm{Max}_v(i)$, which check that the score $v$ is the maximum, and $\mathrm{Rightmost}_{v,c,b}(i)$, which retrieve the value at the rightmost position with score $v$:
$$\mathrm{Max}_v(i) := \blacktriangleright_j [M(i, j), S_{>v}(i, j)] \; 0 : 1$$
$$\mathrm{Rightmost}_{v,c,b}(i) := \blacktriangleright_j [M(i, j), S_v(i, j)] \; V'_{c,b}(j) : 0.$$

Then we can add operations for $\mathrm{Att}_{c,b}(i)$, which hold just in case the $b$-th bit of the $c$-th coordinate of the attention output is 1, by taking a disjunction over the finitely many possible scores:
$$\mathrm{Att}_{c,b}(i) := \bigvee_{v \in \mathbb{F}} (\mathrm{Max}_v(i) \wedge \mathrm{Rightmost}_{v,c,b}(i)).$$

*Smaller version:*

We need to define a predicate $\mathrm{Argmax}(i, j)$ that tests whether $j$ maximizes $S(i, j)$. To do this, we define a sequence of Boolean vectors that test whether $j$ maximizes bits $b, \dots, B - 1$ of $S(i, j)$:
$$\mathrm{Argmax}_B(i, j) = 1$$
For $b = B - 1, B - 2, \dots, 0$:
$$\mathrm{Max}_b(i) := \blacktriangleright_j \left[ M(i, j), \mathrm{Argmax}_{b+1}(i, j) \wedge S'_b(i, j) \right] \; 1 : 0$$
$$\mathrm{Argmax}_b(i, j) = \bigwedge_{b'=b}^{B-1} \left( S'_{b'}(i, j) \leftrightarrow \mathrm{Max}_{b'}(i) \right)$$
$$\mathrm{Argmax}(i, j) = \mathrm{Argmax}_0(i, j).$$
Finally, we add operations that simulate attention:
$$\mathrm{Att}_{c,b}(i) := \blacktriangleright_j [M(i, j), \mathrm{Argmax}(i, j)] \; V'_{c,b}(i, j) : 0.$$
To simulate leftmost-hard attention, simply change $\blacktriangleright$ to $\blacktriangleleft$.

For the position-wise feed-forward network, use Lemma 23. $\qquad\qquad\square$

The last step in the program is to use position-wise operations to simulate $\mathcal{T}$'s output layer, yielding an output Boolean vector $Y$. This completes the proof of Theorem 4.

# D  Proofs for Section 5 (Further Results)

## D.1  Proof of Theorem 7 (position embeddings can be simulated by predicates)

Because $\Theta$ has finite image, Lemma 22 still holds for any masked hard-attention transformer with position embedding $\Theta$. Let $\mathcal{P}_\Theta$ be the collection of predicates that test whether the $b$-th bit of the $c$-th coordinate of $\theta_n(i)$ is set. The proof of equivalence of masked hard-attention transformers with **B-RASP** extends easily to equivalence of masked hard-attention transformers with position embedding $\Theta$ and **B-RASP**$[\mathcal{P}_\Theta]$. When converting a transformer to a **B-RASP**$[\mathcal{P}_\Theta]$ program, we represent each coordinate of $\Theta$ with $B$ predicates from $\mathcal{P}_\Theta$. When converting a **B-RASP**$[\mathcal{P}_\Theta]$ program to a transformer, we represent each predicate in $\mathcal{P}_\Theta$ with its own coordinate, whose value is in $\{0, 1\}$.

Since Theorems 1 and 2 hold for any collection of unary predicate symbols, **B-RASP**$[\mathcal{P}_\Theta]$ is equivalent to **LTL**$[\mathcal{P}_\Theta]$.

## D.2 Proof of Corollary 8 (masked hard-attention transformers with sinusoidal position embeddings recognize the regular languages in $\mathbf{AC}^0$)

Let MOD be the collection of predicates $\mathrm{MOD}^r_m(i)$ for all $0 \le r < m$, which hold just in case $i \equiv r$ (mod $m$).

Let $\Theta$ be a sinusoidal positional embedding. Since the $f_c$ are rational, $\Theta$ has finite image. By Theorem 7, transformers with positional embedding $\Theta$ are equivalent to $\mathbf{LTL}[\mathcal{P}_\Theta]$.

It's easy to see that every predicate in $\mathcal{P}_\Theta$ can be expressed in terms of MOD; for the converse, observe that we can use a 2-layer ReLU network to compute $\mathrm{MOD}^r_m$ (Chiang et al., 2023, Lemma 20):

$$
\begin{aligned}
h(i) &= \mathrm{ReLU}\left(\sin 2\pi r/m \sin 2\pi i/m + \cos 2\pi r/m \cos 2\pi i/m - \cos 2\pi/m\right) \\
&= \mathrm{ReLU}(\cos(2\pi(i-r)/m)) \\
\mathrm{MOD}^r_m(i) &= (1 - \cos 2\pi/m)h(i).
\end{aligned}
$$

Thus transformers with sinusoidal positional embeddings are equivalent to $\mathbf{LTL}[\mathrm{MOD}]$, which is equivalent to $\mathbf{FO}[<, \mathrm{MOD}]$ (Kamp, 1968), which defines exactly the class of regular languages in $\mathbf{AC}^0$ (Barrington et al., 1992).

## D.3 Details for Section 5.4 (depth hierarchy)

### D.3.1 Multi-head attention

To prove Theorem 10 and related results, we need to make Theorem 3 more efficient in terms of the depth of the constructed transformer. To do this, we'll need to make use of multi-head attention. This allows multiple self-attentions at the same depth to be run in parallel. In a multi-head masked hard-attention transformer transformer layer, the equation for the self-attention (Equation (1)) is replaced by

$$
(c_1, \dots, c_n) = \sum_{h=1}^{H} att.h(x_1, \dots, x_n) + (x_1, \dots, x_n)
$$

where each $att.h$ is a self-attention layer.

It is straightforward to extend Theorem 4 to multi-head masked hard-attention transformers, simulating a multi-head masked hard-attention transformer of depth $k$ with a **B-RASP** program of depth $k$. Each head at depth $k$ can be simulated by a **B-RASP** attention operation of attention depth $k$, and their sum can be simulated by a position-wise operation (Lemma 23).

### D.3.2 Parallel composition

The parallelization is accomplished by the following construction.

**Lemma 25.** *A transformer $\mathcal{T}_1$ of depth $k_1$ with $H_1$ heads and a transformer $\mathcal{T}_2$ of depth $k_2$ with $H_2$ heads can be parallel-composed into a transformer $\mathcal{T}_1 \oplus \mathcal{T}_2$ of depth $\max(k_1, k_2)$ with $H_1 + H_2$ heads such that*

$$
(\mathcal{T}_1 \oplus \mathcal{T}_2)(w) = \begin{bmatrix} \mathcal{T}_1(w) \\ \mathcal{T}_2(w) \end{bmatrix}.
$$

*Proof.* First, add layers that compute the identity function to the shallower transformer so that both have depth $\max(k_1, k_2)$.

Next, concatenate their word embedding vectors

$$
(emb_1 \oplus emb_2)(\sigma) = \begin{bmatrix} emb_1(\sigma) \\ emb_2(\sigma) \end{bmatrix}.
$$

At each level, we compose the self-attentions using multiple heads to simulate them in parallel. For each multi-head self-attention layer $att_1$ and $att_2$ at the same depth in each transformer, we use multiple heads to simulate both $att_1$ and $att_2$ in parallel. Let $att_1.h.f_S$ be the score function of

the $h$-th head of $att_1$, and similarly for $att_1.h.M$, $att_1.h.C$, and $att_1.h.f_V$, and similarly for $att_2$. Let $d = d_1 + d_2$ and $H = H_1 + H_2$. Construct a new self-attention layer $att_1 \oplus att_2$ with

$$(att_1 \oplus att_2).h.f_S(x_i, x_j) = \begin{cases} att_1.h.f_S([x_i]_{1:d_1}, [x_j]_{d_1+1:d}) & 1 \leq h \leq H_1 \\ att_2.(h - H_1).f_S([x_i]_{d_1+1:d}, [x_j]_{d_1+1:d}) & H_1 + 1 \leq h \leq H \end{cases}$$

$$(att_1 \oplus att_2).h.M = \begin{cases} att_1.h.M & 1 \leq h \leq H_1 \\ att_2.(h - H_1).M & H_1 + 1 \leq h \leq H \end{cases}$$

$$(att_1 \oplus att_2).h.C = \begin{cases} att_1.h.C & 1 \leq h \leq H_1 \\ att_2.(h - H_1).C & H_1 + 1 \leq h \leq H \end{cases}$$

$$(att_1 \oplus att_2).h.f_V(x) = \begin{cases} \begin{bmatrix} att_1.h.f_V(x_{1:d_1}) \\ \mathbf{0}^{d_2} \end{bmatrix} & 1 \leq h \leq H_1 \\ \begin{bmatrix} \mathbf{0}^{d_1} \\ att_2.(h - H_1).f_V(x_{d_1+1:d}) \end{bmatrix} & H_1 + 1 \leq h \leq H. \end{cases}$$

For the feed-forward networks $ffn_1$ and $ffn_2$, create a new network $ffn_1 \oplus ffn_2$ with

$$(ffn_1 \oplus ffn_2).W^{(1)} = \begin{bmatrix} ffn_1.W^{(1)} & \mathbf{0} \\ \mathbf{0} & ffn_2.W^{(1)} \end{bmatrix} \qquad (ffn_1 \oplus ffn_2).b^{(1)} = \begin{bmatrix} ffn_1.b^{(1)} \\ ffn_2.b^{(1)} \end{bmatrix}$$

$$(ffn_1 \oplus ffn_2).W^{(2)} = \begin{bmatrix} ffn_1.W^{(2)} & \mathbf{0} \\ \mathbf{0} & ffn_2.W^{(2)} \end{bmatrix} \qquad (ffn_1 \oplus ffn_2).b^{(2)} = \begin{bmatrix} ffn_1.b^{(2)} \\ ffn_2.b^{(2)} \end{bmatrix}.$$

It is straightforward to verify the correctness of this construction. $\qquad \square$

### D.3.3 B-RASP to masked hard-attention transformers, preserving depth

We give a more efficient version of Theorem 3, which uses parallel composition to optimize the depth of the constructed transformer.

**Lemma 26.** *Let $\mathcal{T}$ be a transformer (without output layer) with width $d$ and depth $k$, and whose activations are in $\{0, 1\}$. For any function $g \colon \{0, 1\}^d \to \{0, 1\}^d$, there is a transformer $(g \circ \mathcal{T})$ with depth $k$ such that, for all $w$ and $i$, $[(g \circ \mathcal{T})(w)]_i = g([\mathcal{T}(w)]_i)$.*

*Proof.* If $k = 0$: $\mathcal{T}$ consists of just an embedding function $emb \colon \Sigma \to \{0, 1\}^d$. Then $g \circ \mathcal{T} = g \circ emb$ is also an embedding function and therefore a depth-0 transformer.

If $k > 0$: Let $f \colon \{0, 1\}^d \to \{0, 1\}^d$ be the top FFN of $\mathcal{T}$. Then $g \circ f$ is also a function $\{0, 1\}^d \to \{0, 1\}^d$ and can therefore be computed by a single FFN, by Lemma 20. $\qquad \square$

**Theorem 27.** *For any B-RASP program $\mathcal{P}$ of depth $k$ that recognizes a language $L \subseteq \Sigma^+$, there is a multi-head masked hard-attention transformer with depth $k$ that recognizes $L$.*

*Proof.* Let $\mathcal{P}$ be any **B-RASP** program. For any operation $P_t(i)$ of $\mathcal{P}$, we say that a transformer $\mathcal{T}_t$ with width $d$ *simulates* $P_t(i)$ if there is a $c \in [d]$ such that, for all $w \in \Sigma^+$,

$$[\mathcal{T}_t(w)]_{i,c} = \begin{cases} 1 & \text{if } w \models P_t(i) \\ 0 & \text{otherwise.} \end{cases}$$

We prove the following statement by induction on $t$: For any operation $P_t(i)$ of $\mathcal{P}$ with depth $k$, there is a multi-head masked hard-attention transformer with depth $k$ that simulates $P_t(i)$.

The base cases are $t \leq |\Sigma|$, where every operation can be simulated by a transformer with depth 0 using one-hot word embeddings, just as in the proof of Theorem 3.

If $t > |\Sigma|$, assume that each previous operation $P_{t'}(i)$ with depth $k'$ can be simulated by a transformer with depth $k'$. We will construct a transformer of depth $k$ that simulates $P_t(i)$.

- If $P_t(i)$ is an attention operation, then its $S$, $V$, and $D$ predicates have depth at most $(k - 1)$ and therefore depend only on operations which can be simulated by transformers of depth $(k - 1)$, by the inductive hypothesis. Parallel-compose all of these into a single transformer

(using Lemma 25) to obtain a transformer $\tilde{T}_t$ of depth $(k-1)$ that simulates all the operations that $P_t(i)$ depends on.

Then, extend $\tilde{T}_t$ to compute the attention operation as in the proof of Theorem 3. Although that construction uses two transformer layers, the first layer's self-attention just computes the identity function, and its FFN can be fused with the last FFN in $\tilde{T}_t$ by Lemma 26. So there exists a transformer $T_t$ of depth $k$ that simulates $P_t(i)$.

- If $P_t(i)$ is a position-wise operation of depth $k$, it may depend on earlier operations that also have depth $k$. By the inductive hypothesis, these can be simulated by transformers of depth $k$. Parallel-compose them to get a transformer $\tilde{T}_t$ of depth $k$, by Lemma 25. Then, the computation of $P_t(i)$ itself can be fused with the last feed-forward network in $\tilde{T}_t$ by Lemma 26. This again results in a transformer $T_t$ of depth $k$ that simulates $P_t(i)$.

Thus, if $P_T$ is the output vector of $\mathcal{P}$ and has depth $k$, then there is a transformer that simulates $P_T$ and has depth $k$. Add an output layer that transforms the output at position $n$ to $+\frac{1}{2}$ if $w \models P_T(n)$ and $-\frac{1}{2}$ otherwise. Then, $T$ recognizes the same language that $\mathcal{P}$ recognizes. □

### D.3.4 Other attention variants

Earlier, we used transformers and **B-RASP** with only future-masked rightmost-hard-attention, and **LTL** with only **since**. Theorem 10 showed that

$$\cdots \subsetneq \quad \mathbf{LTL(since)}_k \quad \subsetneq \quad \mathbf{LTL(since)}_{k+1} \quad \subsetneq \cdots$$
$$\| \qquad\qquad\qquad \|$$
$$\mathbf{B\text{-}RASP}(\blacktriangleright F)_k \quad \mathbf{B\text{-}RASP}(\blacktriangleright F)_{k+1}$$
$$\| \qquad\qquad\qquad \|$$
$$\mathbf{MUHAT}(\blacktriangleright F)_k \quad \mathbf{MUHAT}(\blacktriangleright F)_{k+1}$$

The separating language was $\text{STAIR}_{k+1}$, which is the language over $\Sigma = \{a, b, c\}$ of strings which, after deleting $c$'s, contain $a^{k+1}$ as a substring. We can recognize $\text{STAIR}_k$ with a formula $\varphi_k = 1$ **since** $\gamma_k$, defined as follows:

$$\gamma_1 = Q_a$$
$$\gamma_k = Q_a \wedge (Q_c \text{ **since** } \gamma_{k-1}) \qquad\qquad k > 1.$$

Note the slight deviation from Etessami and Wilke (2000), because their presentation of **LTL** used the **next** and **eventually** operators as well as non-strict **until**′.

Next, we allow both future-masked rightmost and past-masked leftmost attention. We will notate these with $\mathbf{MUHAT}(\blacktriangleright F, \blacktriangleleft P)_k$ and $\mathbf{B\text{-}RASP}(\blacktriangleright F, \blacktriangleleft P)_k$, respectively. In **LTL**, we allow access to both temporal operators; let $\mathbf{LTL}_k$ be the languages definable by formulas of depth $k$.

**Proposition 28.** *Restricted to only rightmost future-masked and leftmost past-masked attention, multi-head masked hard-attention transformers with depth $(k+1)$ are strictly more expressive than multi-head masked hard-attention transformers with depth $k$.*

*Proof.* The constructions in Theorems 1 and 2 preserve depth for rightmost future-masked and leftmost past-masked attention, so $\mathbf{B\text{-}RASP}(\blacktriangleright F, \blacktriangleleft P)_k = \mathbf{LTL}_k$. Moreover, the constructions in Theorems 3 and 4 are identical regardless of attention type, so $\mathbf{MUHAT}(\blacktriangleright F, \blacktriangleleft P)_k = \mathbf{B\text{-}RASP}(\blacktriangleright F, \blacktriangleleft P)_k$. Finally, Etessami and Wilke (2000) show that with access to both temporal operators it is still the case that $\mathbf{LTL}_{k-1} \subsetneq \mathbf{LTL}_k$ (except the separating language is $\text{STAIR}_{2k}$ instead of $\text{STAIR}_k$). Thus, we have

$$\cdots \subsetneq \qquad \mathbf{LTL}_k \qquad \subsetneq \qquad \mathbf{LTL}_{k+1} \qquad \subsetneq \cdots$$
$$\| \qquad\qquad\qquad \|$$
$$\mathbf{B\text{-}RASP}(\blacktriangleright F, \blacktriangleleft P)_k \quad \mathbf{B\text{-}RASP}(\blacktriangleright F, \blacktriangleleft P)_{k+1}$$
$$\| \qquad\qquad\qquad \|$$
$$\mathbf{MUHAT}(\blacktriangleright F, \blacktriangleleft P)_k \quad \mathbf{MUHAT}(\blacktriangleright F, \blacktriangleleft P)_{k+1}$$

and in particular, $\mathbf{MUHAT}(\blacktriangleright F, \blacktriangleleft P)_k \subsetneq \mathbf{MUHAT}(\blacktriangleright F, \blacktriangleleft P)_{k+1}$. □

Finally, we allow all six types of attention. These will be notated with $\mathbf{MUHAT}_k$ and $\mathbf{B\text{-}RASP}_k$.

**Proposition 29.** *With access to future-, past-, and no masking and both leftmost-hard and rightmost-hard attention, multi-head masked hard-attention transformers of depth $(2k + 1)$ are strictly more expressive than multi-head masked hard-attention transformers of depth $k$.*

*Proof.* As above, $\mathbf{MUHAT}_k = \mathbf{B\text{-}RASP}_k$. However, in the proof of Theorem 2, the simulations of leftmost future-masked and rightmost past-masked attention require two levels of nesting of **since** and **until**, so it only shows that $\mathbf{B\text{-}RASP}_k \subseteq \mathbf{LTL}_{2k}$. As in the previous proof, $\mathbf{LTL}_{2k} \subsetneq \mathbf{LTL}_{2k+1}$ (Etessami and Wilke, 2000). Finally, by Theorem 1, we again have $\mathbf{LTL}_{2k+1} \subseteq \mathbf{B\text{-}RASP}_{2k+1}$. Using all these observations, we conclude that:

$$
\begin{array}{ccccc}
\cdots \subsetneq & \mathbf{LTL}_{2k} & \subsetneq & \mathbf{LTL}_{2k+1} & \subsetneq \cdots \\
 & \cup\mathsf{I} & & \mathsf{I}\cap & \\
 & \mathbf{B\text{-}RASP}_k & & \mathbf{B\text{-}RASP}_{2k+1} & \\
 & \| & & \| & \\
 & \mathbf{MUHAT}_k & & \mathbf{MUHAT}_{2k+1} &
\end{array}
$$

Thus $\mathbf{MUHAT}_k \subsetneq \mathbf{MUHAT}_{2k+1}$. $\qquad\square$

