# OpenReview forum: "Masked Hard-Attention Transformers Recognize Exactly the Star-Free Languages"
_NeurIPS.cc/2024/Conference — NeurIPS 2024 poster_

### Official Review · Reviewer_rKKi · 2024-06-24

**Soundness:** 3
**Presentation:** 2
**Contribution:** 2
**Rating:** 6
**Confidence:** 4

**Summary:**

The paper presents results about several forms of masked unique hard-attention transformers by classifying the class of languages recognized by these models using formal languages. Roughly put, a masked unique hard-attention transformer is an encoder that uses unique hardmax as its attention mechanism. In cases of ambiguity, either the leftmost or the rightmost position is attended to. Additionally, the attention heads are masked, meaning that a position only attends to (strictly) preceding ones (future masking) or (strictly) succeeding ones (past masking).

In summary, the presented results are:
- Future-masked rightmost-hard attention transformers without positional embedding recognize exactly the star-free languages (Theorem 5).
- Non-strict masked hard-attention transformers without positional embedding recognize exactly the stutter-invariant star-free languages (Theorem 6).

To establish these results, the paper introduces an auxiliary language called B-RASP to analyze masked hard-attention architectures. The paper shows that B-RASP exactly captures the considered transformer models (Theorems 3 and 4) and that it is as expressive as strict or non-strict Linear Temporal Logic (LTL) (Theorems 1 and 2). Then, the general proof idea of Theorems 5 and 6 is to translate results from strict or non-strict LTL via B-RASP to the respective transformer model. Additionally, the authors also establish that masked-hard attention transformer recognize the same languages as LTL with monadic predicates.

**Strengths:**

The results, mainly Theorem 5 and 6, in this paper follow an intriguing line of research trying to understand the expressive capabilities of transformer models from a formal languages perspective and, therefore, this research is well placed.

In general, the technical proofs of the appendix are convincing and sufficiently argued and, thus, allow to verify them.

**Weaknesses:**

The contribution of this work is intriguing, though its significance may not be immediately apparent. The primary contributions are Theorems 5 and 6, and the transformer architectures considered are quite specific, possibly tailored for these particular results. There is some uncertainty about their broader interest. As the authors note, masked attention is almost exclusively used in the decoder parts of transformers. However, in this work, it is extensively used in encoder-only transformers.

The presentation of the paper is mixed. The first 5 pages are primarily devoted to preliminaries and the introduction of B-RASP, which, while interesting, serves mainly as a tool. This allocation of space would be justified if B-RASP were particularly complex, but it is relatively straightforward. Consequently, almost all theorems in the main paper lack proof sketches or intuitive explanations. Although the appendix adequately supports the theorems’ statements, the main paper itself does not. Additionally, the paper features a very brief introduction and lacks sections like ‘outlook’, ‘open questions’, ‘limitations’, or an equivalent closing and summarizing section. This omission makes it challenging for readers to place the results in the broader context of research.

The clarity of the paper is also mixed. First, the paper uses a form of LTL with a strict until operator and no next operator. Most readers are likely more familiar with non-strict until and the inclusion of the next operator. While this is a minor issue, a single sentence clarifying this would assist non-expert readers. There are more significant issues, which I address in my questions below. For example, in the brief sketch of Theorem 4, it is stated that the transformer “… can be represented using $\mathcal{O}(1)$ bits.” It is unclear why this result focuses on representation sizes. Another example is Theorem 5, which states that future-only masking transformers recognize exactly the star-free languages. The proof relies on a translation from B-RASP to the transformer, and in this translation, it appears they use a self-attention mechanism requiring mixed masking. This may be a case of unclear explanation, but the current version of the paper makes it difficult to understand. If not clarified, this result could be incorrect.

Edit: My concern about the correctness were cleared in the rebuttal. I changed my rating accordingly.

**Questions:**

- Can you explain in more detail why masked attention is of interest in the encoder parts of transformers?
- In the proof of Theorem 3, Lemma 20 establishes a translation from B-RASP to masked hard-attention transformers. It appears that you use non-strict masking in the first layer to achieve self-attention. Am I correct? If so, how can you prove Theorem 5 based on this, since it concerns future-only masked hard-attention transformers?
- In the proof sketch of Theorem 4, can you elaborate on why you focus on precision?
- Can you explain in more detail the idea presented in line 264 regarding how a specific symbol BOS helps?
- Can you explain in more detail how attention masking helps in recognizing all LTL[Mon] languages, as discussed in the paragraph starting on line 308?

**Limitations:**

The authors clearly describe their considered transformer model and specify where it differs from those commonly used in practice. However, these statements are somewhat scattered throughout the paper. A dedicated subsection summarizing all relevant limitations would be very helpful.

---

> ### Author Rebuttal · Authors · 2024-08-03
>
> Thank you very much for your review. We are glad that you find our work convincing, intriguing, and well-placed.
>
> On B-RASP, please see our global response.
>
> > [T]he paper...makes it challenging for readers to place the results in the broader context of research.
>
> This point is well-taken, and we'll be sure to expand on this given extra space in the final version. Although there are some interesting remaining open questions about unique hard attention transformers, we primarily see these results as showcasing what kinds of results we can hope to obtain in the future: in particular, how do attention masking, position embeddings, and network depth affect expressivity? We've given rigorous answers for unique-hard attention transformers, and hope to obtain answers for more realistic (i.e., soft attention) transformers.
>
> Regarding LTL with strict "since" versus non-strict "since" and "previous": We'll be happy to include a brief remark about this.
>
> > [T]he transformer architectures considered are quite specific, possibly tailored for these particular results.
>
> Our results, which consider strict and non-strict masking (Thm 6), masking direction (Thm 5), and position embeddings ranging from nothing to sinusoidal to Mon (Thm 7, Cor 8-9), run the gamut of masked hard-attention architectures, both common and uncommon. It's true that our main result concerns an unusual combination of strict masking and position embeddings, but this is because it corresponds to the most well-known language class (star-free regular languages). (It may be worth noting that transformers with masking but no position embedding have been tried in practice and perform rather well (e.g., [Haviv et al 2022], [Kazemnejad et al 2023].))
>
> > Can you explain in more detail why masked attention is of interest in the encoder parts of transformers?
>
> It's true that masking is more commonly associated with decoders than encoders. We call the transformers studied here "encoders" because the string is the input rather than the output. But if one wishes, one could view them as decoders, with the string as the prompt and the accept/reject decision as the single-symbol output. On this view, perhaps, the use of future masking seems more natural.
>
> > In the proof of Theorem 3, Lemma 20...It appears that you use non-strict masking in the first layer to achieve self-attention. Am I correct?
>
> We think you are referring line 629, which says: "The first self-attention simply computes the identity function". This means that the self-attention uses zero vectors for the values, letting the residual connection compute the identity function. The masking in this self-attention does not matter. We'll be sure to clarify this in the final version.
>
> On the other hand, if you're referring to the fact that whatever masking is used in the B-RASP program is preserved in the transformer (line 631), then it's true that the transformer uses a mix of masks. But it's explained under Theorem 5 (line 233) that in LTL, any star-free language can be defined using only strict "since," which translates to only strict future-masking in B-RASP, which translates to only strict future-masking in the transformer.
>
> > In the proof sketch of Theorem 4, can you elaborate on why you focus on precision?
>
> Yes, we can elaborate on this in the final version. As explained in the full proof (line 663 and below), this makes it possible to represent any number computed by the transformer using a finite-sized logical formula. We do agree that line 676 is too terse and will expand it in the final version.
>
> > Can you explain in more detail the idea presented in line 264 regarding how a specific symbol BOS helps?
>
> This idea is tangential to our paper, as it concerns soft-attention transformers, but if a string has a prefix $\ell^k$, then at position $k$, a transformer (without position embeddings) cannot count how many $\ell$'s there are; it can only measure what fraction of symbols are $\ell$, which is 100%. But if we add a BOS symbol, then the fraction of symbols becomes $k/(k+1)$, so the transformer can discern different numbers of $\ell$'s.
>
> > Can you explain in more detail how attention masking helps in recognizing all LTL[Mon] languages, as discussed in the paragraph starting on line 308?
>
> It's possible that this paragraph is misplaced and is really more about LTL rather than LTL[Mon].
>
> With neither position embedding nor attention masking, transformers are "permutation equivariant" ([Yun et al 2020]) (insensitive to reordering of the vectors in its input sequence). Position embeddings are one way to break the symmetry, and attention masking is another.
>
> This paragraph makes the further point that even a transformer with a finite-image position embedding and no attention masking would not recognize all languages in LTL. For the final version, we'll reevaluate where the best place to make this point is.
>
> > A dedicated subsection summarizing all relevant limitations would be very helpful.
>
> We'll definitely give this some thought for the final version.
>
> [Haviv et al 2022]: https://arxiv.org/abs/2203.16634
> [Kazemnejad et al 2023]: https://arxiv.org/abs/2305.19466
> [Yun et al 2020]: https://arxiv.org/abs/1912.10077

---

> > ### Comment · Reviewer_rKKi · 2024-08-09
> >
> > Thank you for your comment!
> >
> > I see your point that Lemma 20 uses residual connection instead of a non-strict masked attention. I increase my scoring accordingly and recommend a weak accept, as I am no longer concerned that the results are incorrect.
> >
> > Just a comment: I still feel that the readability of the paper would benefit from a revision of the technical parts (appendix) and presentation of the main parts.

---

> > > ### Author Response · Authors · 2024-08-10
> > >
> > > Thanks very much for your feedback! With all the suggestions given, we can definitely improve the readability of the paper for the final revision.

---

### Official Review · Reviewer_6BGB · 2024-07-11

**Soundness:** 3
**Presentation:** 3
**Contribution:** 3
**Rating:** 6
**Confidence:** 3

**Summary:**

This paper presents new theoretical results related to the expressive power of Transformers. The authors focus on Transformers with "hard" attention (a simplifying assumption) and strict future masking (i.e. attention can only attend to positions to the left). The paper develops an equivalence between such Transformers and B-RASP, and between B-RASP and Linear Temporal Logic (LTL), where B-RASP is a binary-valued version of RASP, a programming language for Transformers proposed by prior work. Thereby, the authors establish an equivalence between the proposed class of Transformers and LTL.

**Strengths:**

* The paper adds to our understanding of the theoretical expressiveness of Transformers.
* The use of B-RASP as an intermediate representation to establish the equivalence between Transformers and LTL was an interesting approach, and perhaps could inspire future work towards establishing the expressiveness of various Transformer variants.

**Weaknesses:**

* While the authors justify their choice of focusing on "hard attention", this puts some limits on the applicability of their results to real Transformers. However, this is also a common assumption made by prior theoretical work related to the expressiveness of Transformers.
* Prior work has already established that hard attention Transformers can express LTL.
* The main results that extend beyond prior work to establish the *equivalence* with LTL focuses on architectures with strict future masking and without positional encodings. Both choices are a very uncommon configuration for Transformers.

**Questions:**

Questions:

* Is there an example of a commonly used positional encoding scheme with "infinite image"? (section 4.3)
* It seems commonly used schemes for relative position encodings (e.g. https://arxiv.org/abs/1803.02155, https://arxiv.org/abs/1910.10683) could presumably enable expressing attention operations with strict future masking, without requiring this to be an implicit part of the architecture. Is this true? (section 4.3)

Suggestions:

* Nit - perhaps briefly defining terms such as complexity classes AC0 and P on their first mention could help make paper more accessible.
* The main results focus on architectures with strict future masking and without positional encodings. Both choices are a very uncommon configuration for Transformers. However, per the question above, strict future masking is not necessary if using common positional encoding schemes. This is discussed to some degree in section 5.3, but this result could have potentially been mentioned earlier to justify the otherwise somewhat odd focus on strict future masking without positional encodings.
* Section 5.2 mentions that "Soft-attention can measure what fraction of symbols are l". Potentially relevant: the RASP paper also discusses an algorithm for this, termed `selector_width`, and how this can be implemented with a start symbol or without (relying on positional encodings).
* Section 5.3 mentions that positional information only comes from attention masking. Perhaps relevant: https://arxiv.org/abs/2305.19466 shows that absolute and relative positions can be recovered from only future masking (although I believe their proof relies on soft attention).

**Limitations:**

Some limitations are mentioned throughout the paper (e.g. section 2.2), but it might be helpful to have an explicit limitations section.

---

> ### Author Rebuttal · Authors · 2024-08-03
>
> Thank you very much for your review! We're glad you found the paper interesting and see its potential to inspire future work.
>
> > The main results [focus] on architectures with strict future masking and without positional encodings. Both choices are a very uncommon configuration for Transformers.
>
> Our results, which consider strict and non-strict masking (Thm 6), masking direction (Thm 5), and position embeddings ranging from nothing to sinusoidal to Mon (Thm 7, Cor 8-9), run the gamut of masked hard-attention architectures, both common and uncommon. It's true that our main result concerns an unusual combination of strict masking and position embeddings, but this is because it corresponds to the most well-known language class (star-free regular languages). (It may be worth noting that transformers with masking but no position embedding have been tried in practice and perform rather well (e.g., [Haviv et al 2022], [Kazemnejad et al 2023]).)
>
> > Is there an example of a commonly used positional encoding scheme with "infinite image"? (section 4.3)
>
> It depends what you mean by "commonly used". In theoretical papers, yes, it's common to use quantities like 1/(i+1), i, or i² in position embeddings (e.g., [Perez et al 2019]). In practice, many commonly-used embeddings only go up to a fixed maximum position and therefore trivially have finite image, while many (e.g., RoPE, ALiBi) are architectural modifications and not just embeddings from positions to vectors.
>
> > It seems commonly used schemes for relative position encodings (e.g. [Shaw et al 2018], [Raffel et al 2020]) could presumably enable expressing attention operations with strict future masking, without requiring this to be an implicit part of the architecture. Is this true? (section 4.3)
>
> Relative position encodings would be interesting objects of further study. In principle, it seems that they could be used to simulate attention masking, but the encodings of [Shaw et al 2018] and [Raffel et al 2020] only go up to a fixed maximum distance, so they would not be suitable for this purpose.
>
> > [P]erhaps briefly defining terms such as complexity classes AC0 and P on their first mention could help make paper more accessible.
>
> Thanks for the suggestion; we'll do that in the final version.
>
> > [I]t might be helpful to have an explicit limitations section.
>
> We'll definitely give this some thought for the final version.
>
> [Haviv et al 2022]: https://arxiv.org/abs/2203.16634
> [Kazemnejad et al 2023]: https://arxiv.org/abs/2305.19466
> [Perez et al 2019]: https://arxiv.org/abs/1901.03429302.html
> [Raffel et al 2020]: https://arxiv.org/abs/1910.10683
> [Shaw et al 2018]: https://arxiv.org/abs/1803.02155

---

> ### Comment · Reviewer_6BGB · 2024-08-07
>
> Thank you for your response. I confirm my original recommendation to accept.
>
> > Relative position encodings would be interesting objects of further study. In principle, it seems that they could be used to simulate attention masking, but the encodings of Shaw et al 2018 and Raffel et al 2020 only go up to a fixed maximum distance, so they would not be suitable for this purpose.
>
> nit: I believe both approaches use clipping of relative distances to handle, in theory, inputs of unbounded length. It seems a casual attention mask could be implemented with relative distance buckets $[\leq-1, 0, \geq1\]$, and a bias of $-\inf$ for the appropriate buckets, which would be supported by either parameterization.

---

> > ### Author Response · Authors · 2024-08-08
> >
> > Thanks very much for your feedback! We appreciate the additional point about the appropriate relative distance buckets.

---

### Official Review · Reviewer_6xK6 · 2024-07-12

**Soundness:** 4
**Presentation:** 2
**Contribution:** 3
**Rating:** 6
**Confidence:** 5

**Summary:**

The paper studies the expressive power of transformer encoders in terms of their ability to recognize regular languages. The main result in the paper establishes that if such encoders are equipped with hard attention, future masking is permitted, and positional encodings are disallowed, then the languages accepted by this model are precisely the ones that can be defined in linear temporal logic (LTL), which in turn coincides with the star-free languages. Several other results are presented in the paper, but in my view, these are corollaries of the main result and known properties of the relationship between LTL and certain classes of regular languages.

**Strengths:**

The ability of Transformers to recognize languages, under different assumptions (e.g., attention mechanism, presence of decoders, type of positional encoding), is by now an active area of research. But this is one of the few results in the area that presents a precise characterization of an important Transformer model. While the proofs are more or less straightforward, I find the results beautiful and relevant, so I support the acceptance of this paper.

**Weaknesses:**

I find the detour through B-RASP unnecessary. First of all, proofs are not presented in the main body, and hence for someone without access to the appendix, there is no support for the claim that this detour is key. What is it that makes it "key"? Second, I think that a direct translation from LTL to Transformers, and back, is possible. The upper bound only depends on the finiteness of the vectors considered by the Transformers. In turn, for the lower bound, you can use similar techniques to those in Barceló et al, i.e., induction on the structure of formulas (this time by using positional masking as opposed to positional encodings).

**Questions:**

Please explain why the detour through B-RASP is key to your proof and why you have decided to leave this connection in the body of the paper instead of keeping it in the appendix as part of the main proof. What kind of conceptual benefit do you feel that this connection brings to the paper?

**Limitations:**

Limitations are correctly addressed in the paper.

---

> ### Author Rebuttal · Authors · 2024-08-03
>
> Thank you very much for your review, and for your assessment of our results as "beautiful" (!).
>
> It's true that the equivalence of LTL and masked hard-attention transformers could be proven directly, without going through B-RASP. We have worked out the LTL to transformer direction outside of this paper. However, we forsee that the proof from transformers to LTL would have the same challenges as the proof of transformers to B-RASP, and would be harder to follow.
>
> We would need a version of Lemma 12 for transformers: every unique hard attention transformer is equivalent to one in which the attention queries do not depend on the position. We think the matrix manipulations needed to prove this would be far more difficult than the present Lemma 12.
>
> This is because LTL does not have a syntax for binary predicates, so we must eliminate relations that depend on two positions before doing the translation into LTL. While this works for masked-hard attention transformers due to Lemma 21, the same technique will not apply to other transformer variants. Thus we foresee the proof of transformers into B-RASP, with its binary score predicates S(i,j), as more relevant to future work than the version directly to LTL. By publishing this version of the proof, we hope to provide a template for extensions towards more realistic transformers.
>
> For the general conceptual benefits of B-RASP, please see our global response.

---

> > ### Comment · Reviewer_6xK6 · 2024-08-07
> > **Response**
> >
> > Thanks for your response. I am still unconvinced about the necessity of going through the B-RASP FORMALISM, but this does not affect my general view of the paper. I stand with my current score and I think that this paper would be a valuable contribution to the conference.

---

> > > ### Author Response · Authors · 2024-08-08
> > >
> > > Thanks very much for your feedback!

---

### Official Review · Reviewer_u2XQ · 2024-07-12

**Soundness:** 3
**Presentation:** 4
**Contribution:** 3
**Rating:** 7
**Confidence:** 3

**Summary:**

The authors connect masked hard-attention transformers (with single-head attention) with LTL via a Boolean version of RASP, namely B-RASP. Due to established results on LTL, they show that this type of transformer recognizes exactly the star-free languages.

While unique hard attention transformers have been studied before, the authors also use strict future (and past) masking which essentially means that each position can't attend to itself but only to positions to the left (or right), respectively. RASP is a programming language by Weiss et al that was invented to have a language very close to how transformers work. This paper uses a Boolean version of it and shows equivalence with hard-attention transformer encoders with strict future masking. Linear Temporal Logic (LTL) is a well researched logic which recently has been linked to unique hard attention transformer encoders and shown equivalent in this paper. Some known results lead to further connections to and characterisations of transformers, including some extensions such as the use of position embeddings. The authors also show that increasing the number of attention layers always increases expressive power in multi-head masked hard-attention transformers.

**Strengths:**

The paper is very well-written and I enjoyed the examples for B-RASP which made it very easy to understand both the programming language and the equivalence proofs.
The first 4 sections lead through the results nicely. The last section is a bit more over the place but gives the impression that the authors looked into multiple directions and found numerous possible extensions that follow with little adjustments from their main result.
The discussion of previous work in between the results gives a good overview about similar lines of work and was very fitting.

**Weaknesses:**

First of all, I was missing a discussion about implications of the results and possible future work.
Unique hard attention transformer encoders have been known to be in AC⁰ and have been linked to LTL[Mon] before. In my eyes, the main contribution of this paper is therefore the result that this characterisation is exact and the results on strict vs. non-strict masking. While these are useful realizations, they might not make the biggest impact on practical applications. Even more so, because hard attentions is hardly used in practice.

There is a typo in line 31: exp(r)essivity

**Questions:**

None.

**Limitations:**

The authors have thoroughly discussed the limitations of their findings one by one.

---

> ### Author Rebuttal · Authors · 2024-08-03
>
> Thank you very much for your review and your positive assessment of our paper!
>
> > In my eyes, the main contribution of this paper is therefore the result that this characterisation is exact and the results on strict vs. non-strict masking.
>
> Yes, but we would also remind the reviewer of the other results in Section 5, which consider not only strict and non-strict masking (Thm 6) but also masking direction (Thm 5), position embeddings (Thm 7, Cor 8-9), and network depth (Thm 10). Put together, these results run the gamut of masked hard-attention architectures, and form a fuller picture with the two results you highlighted.
>
> > I was missing a discussion about implications of the results and possible future work.
>
> This point is well-taken, and we'll be sure to expand on this given extra space in the final version. Although there are some interesting remaining open questions about unique hard attention transformers, we primarily see these results as showcasing what kinds of results we can hope to obtain in the future: in particular, how do attention masking, position embeddings, and network depth affect expressivity? We've given rigorous answers for unique-hard attention transformers, and hope to obtain answers for more realistic (i.e., soft attention) transformers.

---

> > ### Comment · Reviewer_u2XQ · 2024-08-12
> > **Response**
> >
> > Thank you for your response. I am satisfied by the comments and retain my score.

---

> > > ### Author Response · Authors · 2024-08-12
> > >
> > > Thanks very much for your consideration and feedback!

---

### Author Response · Authors · 2024-08-03
**Motivating B-RASP**

RASP [Weiss et al 2021] was designed as a programming language that would help people express transformer computation at a higher level of abstraction than specifying query, key, and value matrices for every attention operation.  It has been quite successful in that regard, though somewhat overpowered -- only some restrictions of it have been implemented so far.

B-RASP was designed to simplify the RASP model of computation down as far as possible while retaining the essence of transformer attention behavior: only binary values and no index vector, where relative position information is available only through future and past masking.  In this context, neither softmax attention nor average attention make sense, and the previously-studied unique hard attention was chosen.

Of course, the choice of hard attention immediately restricts the recognizable languages to those in AC⁰, but the question is to characterize the resulting class of recognizable languages, which we have found is exactly the star-free regular languages.

We also discovered a close relationship with linear temporal logic: future-masked rightmost hard attention is very similar to a since operator in LTL, and past-masked leftmost hard attention resembles an until operator.  This connection enables us to prove a proper hierarchy based on the depth of transformers in Section 5.4, which is an open question for more realistic variants of transformers.

However, we think B-RASP is a more intuitive reflection of transformer computation than formulas of LTL. One key difference is that the score predicate in B-RASP (as in transformers) is a function of two positions i and j, while the value of each LTL formula is defined with respect to one position j.  To make the formal connection, we prove Lemma 12, which shows that in B-RASP score predicates can be made unary, at the cost of possibly an exponential increase in the size of the program.

Another appealing property of B-RASP is that it (like RASP) is formulated as a programming language that can be (and has already been) extended to provide incrementally more computational power.  B-RASP may be more accessible to newcomers than either LTL or transformers, occupying a middle ground between them.


[Weiss et al 2021]: https://arxiv.org/abs/2106.06981

---

### Decision · Program_Chairs · 2024-09-25

**Decision:**

Accept (poster)

**Comment:**

All four reviewers are positive about the contribution and the significance of this paper, and most praised its technical achievement.  On the other hand, none of them recommended acceptance clearly and with high confidence, most had reservations around practical relevance of this work, and some questioned the value of working through the intermediate formalism.